



# Measuring sea ice concentration in the Arctic Ocean using SMOS

Carolina Gabarro[1], Antonio Turiel[1], Pedro Elosegui[1,2], Joaquim A. Pla-Resina[1], and Marcos Portabella[1]

[1]Barcelona Expert Center. Institute of Marine Sciences, CSIC, Passeig Maritim Barceloneta 39 Barcelona, Spain.
[2]Massachusetts Institute of Technology, Haystack Observatory, Westford, MA, USA.

*Correspondence to:* Carolina Gabarro (cgabarro@icm.csic.es)

**Abstract.**

We present a new method to estimate sea ice concentration in the Arctic Ocean using brightness temperature observations from the Soil Moisture Ocean Salinity (SMOS) interferometric satellite. The method, which employs a Maximum Likelihood Estimator (MLE), exploits the marked difference in radiative properties between sea ice and seawater, in particular when observed over the wide range of satellite viewing angles afforded by SMOS. Observations at L-band frequencies such as those from SMOS (i.e., 1.4 GHz, or equivalently 21-cm wavelength) are advantageous to remote sensing of sea ice because the atmosphere is virtually transparent at that frequency. We find that sea ice concentration is well determined (correlations of about 0.75) as compared to estimates from other sensors such as the Special Sensor Microwave/Imager (SSM/I and SSMIS). We also find that the efficacy of the method decreases under thin sea ice conditions (ice thickness $\lesssim$0.6 m). This result is expected because thin ice is partially transparent at L-band thus causing sea ice concentration to be underestimated. We therefore argue that SMOS estimates can be a compelling complement to estimates of ice concentration of both thick and thin sea ice from other satellite sensors such as the Advanced Microwave Scanning Radiometer (AMSR-E and AMSR-2) or SSMIS, enabling a synergistic monitoring of pan-Arctic sea ice conditions.

## 1 Introduction

The Arctic Ocean is under profound transformation. The rapid decline in Arctic sea ice extent and volume that is both observed and modeled (e.g., Comiso, 2012; Stroeve et al., 2012) may have become the key illustration of change in a warming planet, but change is widespread across all Arctic systems (e.g., AMAP, 2012; IPCC, 2013; SEARCH, 2013). A retreating Arctic ice cover has a marked impact on regional and global climate, and vice versa, through a large number of feedback mechanisms and interactions with the climate system (e.g., Holland and Bitz, 2003; Cohen et al., 2014; Vihma, 2014). Despite its importance, our understanding of the evolution of Arctic sea ice decline is incomplete, largely due to a paucity of observations. What is missing is a quasi-continuous, Arctic-wide, comprehensive sea ice observing system.

The launch of the Soil Moisture Ocean Salinity (SMOS) satellite, in 2009, marked the dawn of a new type of space-based microwave imaging sensor. Originally conceived to map geophysical parameters of both hydrological and oceanographic interest (e.g., Martin-Neira et al., 2002; Mecklenburg et al., 2009), SMOS is also making serious inroads in the cryospheric sciences (e.g., Kaleschke et al., 2010, 2012; Huntemann et al., 2014). Developed by the European Space Agency (ESA), SMOS single payload is an L-band (1.4 GHz, or 21-cm wavelength) passive interferometric radiometer that measures the electromagnetic



radiation emitted by Earth's surface. The observed brightness temperature ($T_B$) can be related to moisture content in the soil and to salinity in the ocean surface (Kerr et al., 2010; Font et al., 2013), as can be used to infer sea ice thickness (Kaleschke et al., 2012) and snow thickness (Maaß et al., 2015; Maaß, 2013).

Sea ice concentration (SIC), defined as the fraction of ice relative to the total area at a given ocean location, is often used to

determine other important climate variables such as ice extension and ice volume. SIC has therefore been the target of satellite-based passive microwave sensors such as AMSR-2 and SSMIS since more than 30 years because the brightess temperature of sea ice and seawater are quite distinct. There exists a variety of algorithms to retrieve SIC from $T_B$ observations tuned to those higher-frequency sensors, that is frequencies between 6–89 GHz  (e.g., Cavalieri et al., 1984; Comiso, 1986; Kaleschke et al., 2001; Markus and Cavalieri, 2000; Smith, 1996; Ramseier, 1991; Shokr et al., 2008). Those algorithms present different

advantages and drawbacks depending on frequency, spatial resolution, atmospheric effects, physical temperature, and others. See for example Ivanova et al. (2015) for a review of a sample of thirteen of those algorithms. Although some authors (e.g., Mills and Heygster, 2011a; Kaleschke et al., 2013) have recently explored the feasibility of SIC determination using an aircraft-mounted L-band radiometer, a method that extends satellite-based SIC retrievals down to L-band (i.e., SMOS) frequencies has been missing. We therefore set out to develop a new method, which we present here.

A significant difference between high-frequency and L-band microwave radiometry is that unlike the former, the ice penetration of the latter is non-negligible (Heygster et al., 2014) . In other words, ice is more transparent (i.e., optically thinner) at low microwave frequencies than at high. As a consequence, the brightness temperature measured by an L-band antenna is not only emitted by the topmost ice surface layer but by a larger range of deeper layers within the ice. Thanks to that increased penetration in sea ice (about 60 cm depending on ice conditions), the SMOS L-band radiometer is also sensitive to ice thickness

(Kaleschke et al., 2012; Huntemann et al., 2014). In fact, ideally one would want to estimate both SIC and sea ice thickness simultaneously (e.g., Mills and Heygster, 2011a), which is left for future work.

Wilheit (1978) analyzed the sensitivity of microwave emissivity to a variety of geophysical variables such as atmospheric water vapor, sea surface temperature, wind speed, and salinity as function of frequency (Figure 1).  The figure illustrates that L-band observations are in a sweet spot, with the effect of all variables but salinity being minimal around the SMOS frequency.

Wilheit (1978) also showed that the signature of multi-year (MY) and first-year (FY) ice overlap in the lower microwave frequencies, while this is not the case at higher frequencies.

We exploit some of SMOS key observational features in this study to develop a new method to estimate SIC. These include  a spatial resolution of about 50 km, combination of acquisition modes involving dual and full polarization, continuous multiangle viewing between nadir and 65°, wide swath of about 1200 km, and 3-day revisit time at the equator but more frequently at the

poles. In particular, the multiangle viewing capability of SMOS is a noteworthy feature; it means that the same location on the Earth's surface can be observed quasi-simultaneously from a continuous range of angles of incidence as the satellite overpasses it.

The new method we present in this paper uses SMOS brightness temperature $T_B$ and a Maximum Likelihood Estimator (MLE) to obtain SIC maps in the Arctic Ocean. We describe SMOS data and a radiative transfer model for sea ice that allows

us to compute its theoretical emissivity,  in Sections 2 and 3. We then introduce the concept of tie points and its sensitivity to



different geophysical parameters to help with SIC retrievals via algorithmic inversion of SMOS data, in Section 4.1, 4.2, 4.3 and 4.4, and the MLE inversion algorithm, in Section 4.5. We then perform an accuracy assessment of SIC estimates using SMOS by comparing them to an independent SIC dataset in Section 5. We close with a discussion and conclusions, in Section 6.

## 2 Data

### 2.1 SMOS data from the Arctic Ocean

Since its launch in 2009 ESA has been generating brightess temperature full polarization data products from SMOS. In this study, we focus on the official SMOS Level 1B (L1B) product version 504 data north of 60° N from 2014 to estimate SIC. (The analysis of the entire SMOS dataset, which continues to be growing, is left for future work). The L1B data contains the Fourier components of $T_B$ at the antenna reference frame (Deimos, 2010), from which one can obtain temporal snapshots of the spatial distribution of $T_B$ (i.e., an interferometric $T_B$ image) by performing an inverse Fourier transform. The $T_B$ data are geo-referenced at an Equal-Area Scalable Earth (EASE) Northern hemisphere grid (Brodzik and Knowles, 2002) of 25 km on the side. The estimated radiometric accuracy of individual $T_B$ measurements from SMOS is ∼2 K at boresight, and can increase up to ∼4.5 K in the Extended Alias Free Field-of-View (Corbella et al., 2011). Proceeding from L1B data, though computationally more demanding than the more traditional L1C data products, has several benefits. For example, it allows one to change the antenna grid from the operational size of 128x128 pixels to 64x64 pixels. As shown by Talone et al. (2015), the smaller grid is optimal in that it helps mitigate some of the spatial correlations between measurements that are present in the larger grid.

We corrected the $T_B$ data for a number of standard contributions such as geomagnetic and ionospheric rotation and atmospheric attenuation (Zine et al., 2008). The galactic reflection is not significant at high latitudes. We then filtered out outliers, and edited out $T_B$ estimates in regions of the field of view that are known to have low accuracy due to aliasing (Camps et al., 2005), Sun reflections, and Sun tails. We then applied a correction for the brightness temperature at the bottom of the atmosphere.

To lower the noise level, we averaged $T_B$ measurements from both the ascending and descending orbits over periods of 3 days, which thus define the time resolution of our SIC maps, over each grid cell, and also averaged acquisitions in incidence angle of 2° intervals. We used a cubic polynomial fit to interpolate $T_B$ to locations that might otherwise not have $T_B$ estimates over the full range of incidence angles.

### 2.2 OSI-SAF and other sea ice data products

We used SIC maps from the database (product version OSI-401a) of the Ocean and Sea Ice Satellite Application Facility (OSI SAF) of the European Organization for the Exploitation of Meteorological Satellites (EUMETSAT).





These are computed from brightness temperature observations from SSMIS at 19 and 37 GHz, are corrected for atmospheric effects using forecasts from the European Center for Medium Range Weather Forecasts (ECMWF), use monthly tie points (see below), are available on polar Stereographic 10-km grid for both polar hemispheres, and include SIC uncertainty estimates (Tonboe et al., 2016). In this study, we used daily SIC maps in the Arctic Ocean from the OSI-SAF northern hemisphere

products from 2014.

We also used SIC estimates from ice charts generated from various sensors by the National Ice Center (Fetterer and Fowler, 2009).

## 3    Theoretical model of sea ice radiation at microwave wavelengths

The goal of this study is to develop a method that allows us to estimate Arctic sea ice concentration at the SMOS observing

frequency. Our approach will be to first describe the theoretical framework for the radiation emitted by sea ice as it pertains to L-band. We will then describe a procedure that is robust when the values of the physical parameters of the model are unknown, which is often the case.

As we discussed in Section 1, passive radiometers measure brightness temperature $T_B$. This can be expressed as

$$T_B = e_s T, \tag{1}$$

where $e_s$ is surface emissivity and $T$ is the physical temperature of the radiation-emitting body layer.

To calculate $e_s$, we will assume a sea ice model consisting of horizontal layers of three media – air, snow, and thick ice. We will use the incoherent approach (i.e., conservation of energy, instead of wave field treatment in the coherent approach) and the radiative transfer equation (e.g., Burke et al., 1979) to compute the net emission from the third and second media (i.e., ice and snow, respectively) into the first medium (i.e., air). The approach is similar to that used by other authors (e.g., Mills and

Heygster, 2011b; Maaß, 2013).

Emissivity $e$ and reflectivity are related by $e = (1 - \Gamma)$. The reflectivity $\Gamma$ (sometimes also called $R$) is the ratio of reflected and incident radiation at the media boundaries for each polarization, and can be calculated using Fresnel equations, which depend on the dielectric constant of the layers. The frequency-dependent dielectric constant of a medium, described in Debye (1929)), is a complex value defined as $\hat{\varepsilon}(f) = \varepsilon^{'}(f) + i\varepsilon^{''}(f)$, where the real part $\varepsilon^{'}$ is related to the electrical energy that can

be stored in the medium, and the imaginary part $\varepsilon^{''}$ is related to the energy dissipated within the medium. Another important quantity is the refraction index $n$, which is also a complex value, and for nonmagnetic materials is $n = \sqrt{\hat{\varepsilon}}$ (e.g., Ulaby et al., 1986). Note that brightness temperature varies linearly with emissivity (Eq. 1), hence also with reflectivity. However, the dependence of reflectivity on the dielectric constant is nonlinear because the latter, in turn, depends nonlinearly on the angles of incidence and transmission at the boundary and on the conductivity of the medium (e.g., Ulaby et al., 1986). The nonlinearity

is an advantageous property for remote sensing that can be exploited by the multi-angle viewing capability of SMOS.

Figure 2 shows the dependence of brightness temperature with angle of incidence for seawater and sea ice, as well as ice overlaid by a dry snow layer (see Eq. 2 below), for standard Arctic temperature and salinity values . Note that the $T_B$ of





seawater is significantly less than that of ice, and that the latter is slightly less than that of snow over ice. Also note the nonlinear dependence of $T_B$ on incidence angle, the difference between horizontally (H) and vertically (V) polarized waves for all three models, and the higher sensitivity of H with incidence angle than V in ice and snow (e.g., Maaß et al., 2015).

We calculate surface emissivity $e_s$ of our three-layer, plane-parallel radiative transfer model (Eq. 2) by propagating to the surface the reflectivity computed at and through the ice-snow and snow-air media boundaries, and making a number of simplifying assumptions. Specifically, the model assumes (a) that the media are isothermal, (b) that the thickness of the ice layer is semi-infinite so that radiation from an underlying fourth layer (i.e., seawater) does not need to be considered, and (c) that the snow layer is of constant thickness $d$, can be characterized by an extinction coefficient $\kappa_e \approx 2\alpha$ (see Eq. 3), has a small single-scattering albedo $a_s < 0.1$, and the magnetic permeability $\mu$ is equal to one (non-magnetic materials) (Chandrasekhar, 1960; Ulaby et al., 1986).

These assumptions are realistic for the spontaneous emission of sea ice that is thicker than about 60 cm at the observing frequency of SMOS, as we discussed in Section 1, since the underlying seawater then makes no contribution to the overall emissivity. (But see below for the case of thin ice.)

To further simplify our approach, we will assume that the snow layer in the model consists of dry snow, which is typical of winter Arctic conditions. Because dry snow is a lossless medium at 1.4GHz, that means that there will be no attenuation in the snow layer, or equivalently $\kappa_e = 0$. However, dry snow still has an effect in emissivity that changes with angle of incidence according to Snell's law (e.g., Maaß et al., 2015). We make this simplifying assumption because water in a wet snow layer will cause attenuation and therefore decrease the total emissivity, but it is rarely possible to obtain meaningful data on the amount of water in wet snow.

The brightness temperature of a thick-ice, dry-snow layered body that is measured at an angle of incidence $\theta$ with respect to the local vertical, polarization $p$, and observing frequency $f$ can be simply calculated by propagating onto the surface the radiation that results from multiple reflections and refractions at the two media boundaries (i.e., ice-snow and snow-air) accounting for the infinite number of reflections between layers as (Burke et al., 1979):

$$
\begin{aligned}
T_B(\theta, p, f) = {}& T_{snow}\left[1 + (1 - \Gamma_{si})\exp^{-\tau} + \Gamma_{si}\exp^{-2\tau}\right](1 - \Gamma_{as}) \\
& + T_{ice}(1 - \Gamma_{as})(1 - \Gamma_{si})\exp^{-\tau} + T_{sky}\Gamma_{as},
\end{aligned}
\tag{2}
$$

where $\Gamma_{as}$ and $\Gamma_{si}$ are the reflectivity at the air-snow and snow-ice boundaries, respectively, and $T_{snow}$ and $T_{ice}$ are the physical temperature in the snow and ice layers, respectively. The term $\tau$ is the attenuation factor and is defined as $\tau = 2\alpha \sec\theta\, d$, where $\alpha$ is the atenuation constant and $d$ the depth of the snow layer. $T_{sky}$ is the temperature of the cosmic background. The dependence of $T_B$ on $\theta$, $p$, and $f$ is embedded in the expressions of $\Gamma$ and $\tau$.

The middle layer atenuation constant $\alpha$, in the case of a low-loss medium ($\varepsilon''/\varepsilon' << 1$) can be expressed as:

$$
\alpha = \frac{\pi f}{c} \frac{\varepsilon''}{\sqrt{\varepsilon'}}
\tag{3}
$$

where $c$ the speed of light. The skin depth is defined as $\delta_s = 1/\alpha$ (m) and characterizes how deep an electromagnetic wave can penetrate into a conducting medium (e.g., Ulaby and Long, 2014).





Following Ulaby and Long (2014), the dry snow permittivity at L-band can be assumed to be constant and equal to $\varepsilon_{snow} = 1.4759$, with negligible imaginary dielectric constant. Therefore, the snow attenuation coefficient $\alpha_{snow}$ is considered zero at this frequency.

To compute the complex dielectric constant of sea ice $\varepsilon_{ice}$ (which is needed to compute $\Gamma_{si}$), we use the classic empirical relationship by Vant et al. (1978). In this model, permittivity depends linearly on the ice brine volume $V_b$ as,

$$\hat{\varepsilon}_{ice} = a_1 + a_2 V_{br} + i(a_3 + a_4 V_{br}) \tag{4}$$

where $V_{br} = 10 V_b$, and the coefficients $a_i$ can be obtained by linear interpolation to 1.4 GHz of the laboratory values from microwave measurements at 1 and 2 GHz (refer to Vant et al. (1978) for coefficient values).

The sea ice brine volume $V_b$, can be computed using Cox and Weeks (1983) as follows:

$$V_b = \frac{\rho S}{F_1(T) - \rho S F_2(T)} \tag{5}$$

where $\rho$, $S$, and $T$ are sea ice density, salinity, and temperature, respectively. The $F$ functions are cubic polynomials derived empirically, namely

$$F_j(T) = \sum_{i=0}^{3} a_{ij} T^i \tag{6}$$

where the values of the coefficient $a_{ij}$ were given in Leppäranta and Manninen (1998) for ice temperatures between –2 °C and 0 °C, and for lower temperatures in Cox and Weeks (1983); see also Thomas and Dieckmann (2003).

We also calculate the theoretical emissivity $e_s$ of a four-layer model using the procedure described above. The additional layer in this model is the seawater under sea ice. The model value necessary for the dielectric constant of seawater comes from Klein and Swift (1977). This layer did not need to be considered for the case of (optically) thick ice described above, but it becomes "visible" for the case of (optically) thin ice (i.e., thicknesses $\lesssim$60 cm, depending on the ice temperature and salinity). Because the emissivity of seawater is significantly less than that of sea ice (Figure 2), the net effect of introducing a seawater-ice boundary in the model is of an overall decrease in surface emissivity, hence $T_B$ (as illustrated in Figure 5), relative to emissivity of thick ice (Shokr and Sinha, 2015).

## 4 Methods

### 4.1 Definition of robust indices from brightness temperature

It is rarely possible to obtain the ancillary geophysical data such as sea ice temperature, salinity, and ice thickness that is required to estimate brightness temperature from the microwave remote sensing model described above. Therefore, making assumptions and approximations becomes critically important as discussed therein. It is possible, however, to define a number of indices combination of brightness temperature observations that are less sensitive to the unknown physical parameters. For example, estimates of soil moisture or sea ice concentration from radiometric measurements are often derived by combining





$T_B$ measurements obtained from different polarizations, frequencies, and angles of incidence (Becker and Choudhury, 1988; Owe et al., 2001). Combinations of $T_B$ measurements might result in lower sensitivity than $T_B$ to the exact physical conditions, but good enough to differences in conditions such as when a phase change occurs, thus increasing robustness.

We will use two indices hereafter, the polarization difference (PD) index and the angular difference (AD) index. The PD
index is defined as the difference between $T_B$ measurements obtained at vertical $T_{B_V}$ and horizontal $T_{B_H}$ polarizations as

$$\text{PD} = T_{B_V} - T_{B_H}. \tag{7}$$

The AD index is defined as the difference between two vertical polarization $T_B$ measurements obtained at two different angles of incidence as

$$\text{AD} = T_{B_V}(\theta + \Delta\theta) - T_{B_V}(\theta). \tag{8}$$

Figures 3 and 4 show the variation of PD and AD for the thick-ice model with angle of incidence, respectively. In defining AD, we use vertical rather than horizontal polarization because identification of the three media is facilitated by the larger dynamic range and non-crossing signatures of the former (Figure 4). We choose $\Delta\theta = 35°$ angle difference because this value represents a good compromise between sensitivity and accuracy in the case of SMOS (Camps et al., 2005) and, importantly, is also well supported by the wide range of satellite viewing angles that characterizes SMOS.

**4.2  Calibration of sea ice concentration using tie points**

Tie points are widely used for retrieving SIC with higher frequency radiometers, as well as in other fields such as photogrammetry (e.g., Khoshelham, 2009). In this study, we will use tie points as ground-truth estimates of sea ice concentration. In this context, tie points are reference values of the two radiometric end-members for ocean pixels in the Arctic, that is pixels that are completed covered by sea ice (i.e., 100% ice concentration) and pixels of open water (i.e., 0% ice concentration). In this ap-
plications, tie points can therefore be viewed as SIC calibration points because their expected radiation can be unambiguously determined. .

Figure 3 shows theoretical PD tie-point values for open water and sea ice, as well as ice with a snow layer. The values for an angle of incidence of $50°$ are marked by solid red circles. This angle represents a good compromise in PD contrast between the two media and SMOS accuracy (Camps et al., 2005). The two bounding values are (seawater) 62.9 K and (ice and snow)
26.8 K. The large difference between tie-point values suggests that it is possible to estimate SIC at L-band.

Figure 4 shows theoretical AD tie-point values for an angle of incidence difference $\Delta\theta = 35°$ and angles of incidence up to $\theta = 30°$ (which, per Eq. 8, represents the $T_{B_V}$ difference between $\theta = 60°$ and $\theta = 25°$). The values for an angle of incidence of $25°$ are marked by solid red circles, for which the tie points are (seawater) 51.8 K and (ice with snow) 8.6 K (see also Table 1). Hereafter, AD and PD will be evaluated at the incidence angles indicated above.

Figure 5 shows that $T_B$ at nadir, computed here using the four-layer radiative transfer model, increases non-linearly as function of ice thickness up to the saturation value of $\sim$250 K, which is reached when ice becomes $\sim$70-cm thick. In the figure, $T_B$ estimates start at an ice thickness of 5 cm because there is a discontinuity in the Burke model as the thickness of ice tends




to zero (e.g., Kaleschke et al., 2010; Mills and Heygster, 2011a; Maaß, 2013; Kaleschke et al., 2013). Also shown in Figure 5 are theoretical AD and PD values for the incidence angles indicated above. Compared with $T_B$, the variation of both AD and PD with ice thickness is significantly smaller.

### 4.3 Sensitivity of sea ice concentration to surface emissivity changes

We now turn to calculate SIC sensitivity to changes in surface emissivity due to variations in the physical properties of sea ice (i.e., salinity, temperature, thickness) for the three $T_B$, PD, and AD indices. This is done following a standard error propagation method (as also used in (Comiso et al., 1997)). It is important to determine how changes in ice conditions, which are rarely available, affect SIC estimates through those three indices to try to minimize SIC errors obtained using SMOS.

Table 2 lists the theoretical sensitivities, given by the partial derivatives, of the ice indices $I$ ($I = T_B$, PD or AD) to the

geophysical variables of ice: physical temperature (i.e., $\delta I/\delta T$), salinity ($\delta I/\delta S$), and thickness ($\delta I/\delta d$). Those sensitivites are calculated using the models in Section 3. In order to assess which index is less sensitive to changes in a given geophysical variable, we will calculate absolute sensitivites, defined as the sensitivies multiplied by the dynamic range of the measurements.

Assuming a linear dependency of SIC on the indices, and knowing the value of the tiepoints of sea ice (SIC=100%) and seawater (SIC=0%), one can compute the slopes of these linear relations for $T_B$, PD, and AD (i.e., $\delta SIC/\delta T_B$, $\delta SIC/\delta PD$ and

$\delta SIC/\delta AD$). From data in Table 1, we obtain the linear slopes as: $\delta SIC/\delta T_B = 0.65$, $\delta SIC/\delta PD = 2.32$, and $\delta SIC/\delta AD = 2.77$. These slopes can be used to propagate $T_B$, AD, and PD errors to SIC errors.

We will assume reasonable values for the variability of the physical parameters $T, S$ and $d$ of ice (generically denoted by $g$), as follows: $\Delta T$=5 K, $\Delta S$=4 psu, and $\Delta d$=30 cm. Using the values in Table 2 and the slopes calculated above, one can finally compute SIC errors associated to the geophysical variability of $g$ when the index $I$ is used to evaluate SIC following Eq (9).

$$\Delta SIC|_g = \left|\frac{\delta SIC}{\delta I}\right| \cdot \left|\frac{\delta I}{\delta g}\right| \cdot \Delta g \qquad (9)$$

To evaluate the final impact of geophysical variability on the SIC evaluation using the index $I$, we compute the root-sum-squared (RSS) of the SIC uncertainties due to the geophysical parameters (Table 3). The table shows that AD is the most robust index to retrieve SIC, slightly better than PD, and significantly better than $T_B$, as $T_B$ is highly sentitive to ice thickness variations. Even given the uncertainties in the theoretical physical model of ice, we consider the differences significant enough

to focus on inversion algorithms using the PD and AD indices, and not $T_B$, as done by other authors (e.g., Mills and Heygster, 2011a)).

### 4.4 Comparison with empirical tie points

Following the theoretical analysis above, we now turn to evaluate its performance empirically. We therefore select several regions of interest in the Arctic Ocean where SIC has been unambiguously determined to be either 0% or 100% by other sensors

and methods. To identify such regions, we used SIC maps from OSI-SAF and from the National Ice Center. In particular, we selected the open seawater region between latitudes 55°–70° N and longitudes 20° W and 25° E, which comprises more than

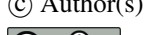



about 2000 pixels in a typical SMOS image. For sea ice, we selected the multi-year (MY) ice region between latitudes 78°–83° N (the northernmost latitude observable by SMOS) and longitudes 75°–150° W, which comprises about 1000 pixels per SMOS image.

We calculated SMOS radiometric values of these target regions to evaluate their potential as empirical tie points for seawater and sea ice. Starting with $T_B$, Figure 6 shows the temporal variation, in 2014, of the spatially averaged (median) $T_B$ at nadir of the two geographic regions above. The values are consistent with the modeled values in Table 1. For the seawater region, the Figure shows that the brightness temperature is constant, at about 99 K, to within $\sim$2.5 K (one $\sigma$ standard deviation) throughout the year. For the ice region, $T_B$ is also stable during the non-summer months, but it drops by about 20 K during the summer season due to changes in surface emissivity associated with snow and ice melt and concurrent formation of meltwater ponds. The factor-two increase in formal error in summer relative to winter is also an indication of increased radiometric variability in surface conditions (as shown in Table 1).

Figure 7 shows that the temporal radiometric stability of the seawater region throughout 2014, and that of sea ice during the non-summer months, is also reflected in the AD and PD indices, as one would expect. This suggest that a different set of tie-points during winter and summer period could be beneficial for the quality of the SIC maps.

Figure 8 shows a 2-D scatter plot of AD and PD indices for the two regions above during the March 2014 (winter tie-point) and during July (summer tie-point). The index values associated with seawater and ice form two well-differentiated clusters, which implies that the two types of regions can be clearly segregated using these indices. This is also true for the summer tie-points eventhough in this case the dispersion is larger and values are closer to sea tie-points, as expected following Figure 7.

Table 1 lists the modeled (with snow and without) and observed TB, AD, and PD tie-point values for winter and summer 2014, and the standard deviation ($\sigma$) of the measurements . It is encouraging that most of the values are in agreement at about $2\sigma$ , despite underlying model assumptions such as uniform sea ice temperature and specular ocean surface. Another important result is that the observed SMOS data is closer to the model when snow is considered.

## 4.5 Retrieval of sea ice concentration algorithm

The brightness temperature of mixed pixels, that is, ocean pixels partially covered by sea ice, can be expressed as a linear combination of the brightness temperature of ice and seawater weighted by the percentage of each surface type (e.g., Comiso et al., 1997):

$$T_{B_{mixed}} = C T_{B_{ice}} + (1-C) T_{B_{water}} \tag{10}$$

where $C$ is the fraction of ice present in a pixel, with $C = 1$ corresponds to 100% of ice and $C = 0$ to 0% of ice, or equivalently 100% of seawater. Since AD and PD (Eqs. 7-8) depend linearly on brightness temperature, Eq. (10) leads to analogous concentration expressions for AD and PD (see below).

There are several possible strategies to estimate sea ice concentration at a given pixel from the AD and PD values measured at that pixel. The simplest approach is to consider that the values of the tie points are good representatives of the values of AD




and PD at the respective medium, i.e., seawater and sea ice, such that

$$\mathrm{AD} \approx C\,\mathrm{AD}_{ice} + (1-C)\,\mathrm{AD}_{water}$$
$$\mathrm{PD} \approx C\,\mathrm{PD}_{ice} + (1-C)\,\mathrm{PD}_{water} \tag{11}$$

Concentration $C$ can thus be retrieved from the value of either AD or PD by inverting the associated linear equation. In general, $C$ can also be evaluated simultaneously with the AD and the PD observations by averaging the values obtained from both indices, as:

$$C = \frac{1}{2}\left[\frac{\mathrm{AD} - \mathrm{AD}_{water}}{\mathrm{AD}_{ice} - \mathrm{AD}_{water}} + \frac{\mathrm{PD} - \mathrm{PD}_{water}}{\mathrm{PD}_{ice} - \mathrm{PD}_{water}}\right]. \tag{12}$$

This is known as the Linear Estimation of SIC. However, this approach might be too simple, as the values of AD and PD on ice and seawater can have some non-negligible dispersion due to geophysical conditions and to radiometric noise.

In this paper, a new inversion algorithm to estimate $C$ is presented, which considers that AD and PD have known distributions, and by combining the observations it is possible to infer the value of $C$ that is statistically more probable, given those observations.

The distributions of the SMOS AD and PD are unimodal and symmetric (not shown), thus allows to approximate them by Gaussians and the pure ice and pure sea measurements are independent. Therefore we can easily use a Maximum-Likelihood Estimation (MLE) approach. The MLE has many optimal properties in statistical inference such as (e.g., Myung, 2003) sufficiency (the complete information about the parameter of interest is contained in the MLE estimator), consistency (the true value of the parameter that generated the data is recovered asymptotically, i.e. for sufficiently large samples), efficiency (asymptotically, it has the lowest-possible variance among all possible parameter estimates), and parameterization invariance (same MLE solution obtained independent of the parametrization used).

Assuming the linearity superposition of indices (Eqs. 11), it follows that the distributions $\rho$ of AD and PD in a general ocean pixel can be expressed as:

$$\rho_{\mathrm{AD}} \sim \mathcal{N}\left(C\,\overline{\mathrm{AD}}_{ice} + (1-C)\,\overline{\mathrm{AD}}_{water},\ \sqrt{C^2\,\sigma^2_{\mathrm{AD}_{ice}} + (1-C)^2\,\sigma^2_{\mathrm{AD}_{water}}}\right) \tag{13}$$
$$\rho_{\mathrm{PD}} \sim \mathcal{N}\left(C\,\overline{\mathrm{PD}}_{ice} + (1-C)\,\overline{\mathrm{PD}}_{water},\ \sqrt{C^2\,\sigma^2_{\mathrm{PD}_{ice}} + (1-C)^2\,\sigma^2_{\mathrm{PD}_{water}}}\right) \tag{14}$$

where the bar over the AD and PD indices refers to their mean values, the subindex identifies the medium, and $\sigma$ is the associated standard deviation for each index and media. To obtain mean and standard deviation values, we used the SMOS measurements at the tie-point regions and periods discussed above (see Sect. 4.4). The symbol $\mathcal{N}$ means normal probability density function, that is:

$$f(x|\mu,\sigma) = \frac{1}{\sigma\sqrt{2\pi}} \cdot e^{-\frac{(x-\mu)^2}{2\sigma^2}} \tag{15}$$

As a first approximation, we will consider AD and PD two independent variables. It thus follows that the likelihood function $\mathcal{L}$ is equal to the product of their distributions or, equivalently and conveniently, to their sum when expressed with their



logarithms ln, as:

$$\widehat{l} = \ln(\mathcal{L}) = \ln(\rho_{AD}) + \ln(\rho_{PD}) \tag{16}$$

The MLE of SIC is the value of $C$ that maximizes the log-likelihood function $\widehat{l}$. The MLE of SIC is less prone to errors and integrates in a more natural way the information from both indices.

## 5  Results

### 5.1  Internal consistency of SMOS SIC retrievals

We calculated AD and PD index values from the brightness temperature data from SMOS and used the MLE approach described above to obtain SIC estimates over the Arctic Ocean, in 2014. We estimated SIC using, for seawater tie points, a single, year-round median value and associated standard deviation for each index and, for ice tie points, two sets of values, as suggested by the results in Figure 7. For the first set, we used for all epochs the median of the tie-points values between December and May (see Table 1 ), i.e., the winter months when Arctic sea ice extension is maximum. For the second set, we used those same winter values for the months of October through May but the average of the summer values for the months between June and September (see Table 1 ). We did not use the October nor November data to compute ice tie-points values because these are epochs of maximum extension of thin ice, which are known to cause biases in SIC estimates at the SMOS working frequency.

Figure 9 shows the root-mean-square (RMS) error, relative to OSI-SAF, of SIC retrievals over the Arctic Ocean using  four types of inversion methods retrievals and two sets of tie-points values. Using a set of summer tie-points (black plain line) values reduces the RMS error with respect using only one unique tie-point for the whole year (black dotted line). The RMS reduction is about 24% and 12% in July and August, respectively, and to smaller degree in June and September. Therefore, we will hereafter use a different sets of tie-points values in summer and winter.

Furthermore, using the set of summer-winter tie-points bi-values, Figure 9 also shows the RMS error, relative to OSI-SAF, for four types of inversion methods. These consists of the four combinations that result from the two inversion algorithms described above, namely linear and MLE, and the two indices whereby only one, AD, or both, AD and PD, are used for each retrieval. The lowest RMS values through all months in 2014 but January are obtained with the MLE inversion algorithm and the AD index. This RMS pattern is closely mimicked by the linear method, but at ∼5-10% increased noise level. Larger RMS values and increased temporal variability can be observed when the PD index is also used. The RMS error of all retrievals is largest in the Fall, in particular if the PD index is used. Those are months of ice formation, therefore vast regions become covered with frazil ice, nilas, and thin young ice, following the minimum ice extension of September. All methods converge to similar results in September, since this period is the one with minimum ice extension and minimum thin ice is expected (so very small difference between using AD or AD and PD methods). In the next subsection, a physical explanation and analysis of this Fall behaviour is committed.

Figure 10 shows the spatial variation of the difference in MLE SIC retrievals during 2–5 November 2014 between using only the AD index and using the AD and PD indices (i.e., blue and black lines, respectively, in Figure 9). As expected, the largest



differences are associated with regions of thin ice formation, in particular in the Laptev Sea, Kara Sea, and along the edge of the ice pack both in the western Arctic and the Atlantic sector. Together, the spatio-temporal snapshots in Figures 9–10 highlight the sensitivity of PD to the presence of thin ice and its effect on SIC retrieval errors. This conclusion is not fully consistent with the anlysis done in Section 4.3, on the threoretical dependence of the indices ($T_B$, PD, AD) to ice thickness. Table 2 shows that,

theoretically, PD is slightly less sensitive to thin ice than AD. However, the AD index is the one less sensitive (lowest RSS) to variations of all the analysed variables. Therefore, we will hereafter use the AD index, summer-winter tie-points values, and an MLE-based estimator for SIC retrievals.

## 5.2 Accuracy assessment of SMOS SIC retrievals

Having evaluated the internal consistency of the SMOS SIC retrievals, and in the process chosen a minimum-error approach,

we now turn to evaluate the accuracy of those retrievals. Although there does not exist a representative (in the space-time domain) ground-truth dataset that allows us to assess the accuracy of the SMOS retrievals, the SIC estimates from OSI-SAF (already used above) are a good option because they are independent from SMOS, the spatio-temporal sampling and resolution of their products is commensurate with SMOS, and their error budget is available.

Figure 11 shows the spatial distribution of SIC in the Arctic Ocean estimated from (a) SMOS for the 3-day period between

2–5 March 2015, (b) OSI-SAF on 4 March 2014, and (c) the difference between (b) and (a), in that order. March is the month of maximum sea ice extent. As was also suggested above, the largest differences between both algorithms are located at the margins of the sea ice cover, where thinner ice can be expected.

Figure 12 is the same as Figure 11 but here for November, the month of maximum extension of thin young ice (see above). Significant differences are now observed over a much wider area of the Arctic Ocean including the Barents, Kara, Laptev, East

Siberian, and Beaufort seas. That is because thin ice is widely present in this season, and the radiometric response of SMOS to thin ice and the response of the microwave radiometers used by OSI-SAF are distinctly different.

That response is linked to the brightness temperature measured by a passive microwave radiometer, which increases with sea ice thickness up to a saturation value. Such an increase is more gradual for low frequencies and horizontal polarization (e.g., Ivanova et al., 2015). At the SMOS L-band, the increase of emissivity with ice thickness reaches saturation for an ice thickness

that is about 60 cm, depending on ice salinity adn temperature (Kaleschke et al., 2012) whereas at the OSI-SAF frequencies is only a few cm. It is reasonable to infer that the observed SIC differences between SMOS and OSI-SAF are mainly associated with the different thickness of thin ice and ensuing penetration depths. For example, for pixels that are 100% covered by thin ice of say 25-cm thickness, the $T_B$ values from SMOS for these pixels will be lower than the tie-point values because these were computed from thick, MY ice. This contrast leads to a difference in classification of such pixels, as a mixed water-ice

pixel in the case of SMOS, but as a 100% ice pixel with OSI-SAF. In other words, SIC of a seas covered by frazil ice and nilas will be higher for OSI-SAF than SMOS.

To further analyze this classification difference, we calculated the probabilities of SMOS SIC conditioned by values of OSI-SAF SIC using a full year, 2014, of Arctic-wide estimates. Figure 13 shows (red) the probability of estimating a SIC value with SMOS that is less or equal than 5% when the estimated OSI-SAF SIC is 0%. As expected, the conditioned probability



is very high throughout the year. This implies that both products have a similar ability to detect (close to) 100% ocean pixels. This implies that the probability of having high SMOS SIC values when OSI-SAF is low is almost zero, which also means that the rate of triggering false alarms on ice detection with SMOS is low.

Figure 13 also shows (blue) the opposite situation, that is the probability of estimating a SMOS SIC equal or higher than 90% while the OSI-SAF SIC is 100%. During the winter period (between January and April), the conditioned probability is notably high (near 0.90). Then it decreases sharply during spring and most notably in summer. The change in conditioned probability starting in the spring could stem from a change in ice properties as well as differences in the retrieval algorithm. In ice properties because as the snow becomes wetter with the onset of the melt season in the spring, the observed radiometric signature will start to change. And in algorithms because OSI-SAF uses dynamically-adjusted tie points (every 30 days) while SMOS uses two tie-points values (summer and winter), which would explain the decrease of the conditioned probability. The observed increse of the condicioned probability in June could be due to the use of summer tie-point (applied from June to September) which improve the RMS with respect OSISAF as shown in Figure 9. The low conditioned probability in Fall can be explained by the presence of thin ice, as described above.

Figures 14 map the spatial distribution of the conditioned probability of SIC estimates for the months of March (a) and November (b). In the figures, the Arctic Ocean has been color-coded in three regions whereby (red) both products have SIC above 0.9, (light blue) OSI-SAF SIC is more than 0.9 while SMOS SIC is less than 0.9, and (dark blue) OSI-SAF and SMOS SIC is less than 0.9. It becomes apparent that the light blue regions outline the edge of the ice cover, thus in good correspondence with the expected areas of thin ice.

Figure 15 shows the monthly spatial coefficients of determination (that is, the square of the correlation coefficient) between SMOS and OSI-SAF SIC throughout 2014. Because the values of SIC tend to be either 0 or 1 over wide Arctic regions, we have excluded both extremes from the figure as this would lead to too high, non-significant values of correlation. Thus, we have only included SIC values between 0.05 (5%) and 0.95 (95%) when computing correlations. During the winter months, the correlation is high (more than 0.65), what again is consistent with our interpretation about the role of thin ice in SMOS SIC (during winter thin ice is scarce and is present only at the edge of the ice cover). As melt starts, the correlation between SIC estimates continues to be high, thanks to the summer tie-point. In September, ice cover extent is at minimum but because ice growth has not started yet there is almost no thin ice, and the correlation remains high. The correlation drops in the Fall (between October and December) because ice growth starts by freezing of the sea surface, producing large amounts of new thin ice.

# 6 Discussion and Conclusions

According to Ivanova et al. (2015) the first source of error in the computation of sea ice concentration from higher frequency radiometers is the sensitivity to changes in the physical temperature of sea ice. The atmosphere has been identified as the second source of error, especially for the presence of water vapor and cloud liquid water. Another problem faced by higher frequency radiometers is that the SIC retrievals are affected by the thickness of snow cover, which is difficult to determine.





However, the sensitive of the brightness temperature to sea surface temperature, atmosphere, and wind speed are clearly reduced when observing the sea surface with radiometers working at lower frequencies(Figure 1), thus making SMOS more reliable in those situations. Moreover, SMOS SIC is not affected by the snow thickness as stated in Section 3.

Ivanova et al. (2015) states that the observed time trends in the measurements obtained by higher frequency radiometers are not only caused by trends in sea ice extent, but also by trends in the atmospheric and surface effects influencing the microwave emission measured by the satellite. Those authors observed seasonal changes on the ice tie-point of up to 10 K seasonally. In order to compensate those effects, they propose to dynamically derive the tie-points using a two-week running window; therefore, a new set of tie points is defined daily.

Figures 6 and 7 show that SMOS $T_B$, PD, and AD have low sensitivity to surface physical changes, and present small trend (for the year observed) . Thanks to that, one can safely assume two sets of static (i.e., not temporally varying) tie-points (summer and winter) for SMOS data. The advantage of the static tie-points is that we can average over a large data record, thus substantially reducing the uncertainty on the empirically derived values.

. On the other hand, the spatial resolution of SIC measurements with SMOS is about 35 km, which is low compared to the ∼3-km resolution that can be achieved with higher-frequency radiometers. Therefore, SMOS-based SIC estimates may be better suited for global climate studies. Another problem that the SMOS SIC maps have to deal is the underestimation of SIC values when thin ice (less than ∼0.60 m) is present, which are characteristic of the ice edges and freeze-up periods.

Two indices derived from brightness temperatures, the Polarization Difference (PD) and the Angular Difference (AD), have been designed to maximize their response to changes in the physical media and to have a low response to changes in the geophysical characteristics of the media, by using theoretical models and sensitivity analysis.

Tie-points, defined as the characteristic values of our reference indices on the different media, have been calculated from SMOS data. When compared to the theoretical values derived from the model some small discrepancies (around 10-20%) have been observed, probably due to the simplifying assumptions (i.e flat surface ice, flat sea, constant temperature at the layers, etc.) used in theoretical models. We have thus decided to follow a more empirical approach. The usage of two sets of tie-points, one for summer and one for winter measurements improve the results of the summer SIC maps, with respect a static unique tie-point. This improvement is not caused by changes in the ice or sea physical temperature, but most probably to the changes in the ice properties, because as the snow and ice becomes wetter during the melt season, the observed radiometric signature change. This effect is also observed on measurements from higher frequency radiometers.

We have introduced the MLE inversion algorithm, to retrieve SIC from SMOS data. The method is based on the maximization of the a posteriori likelihood of the joint distribution of AD and PD, assuming that they are independent and Gaussian distributed. This MLE algorithm is more robust (less noisy) and allows to improve the retrieved SMOS SIC data with respect a linear inversion method, since the former takes into account the dispersion (error) of the tie-points (reference) data, which makes the algorithm more robust to noise on $T_B$. Better quality maps results when using only AD index; The usage of PD in addition to AD is in detriment to the SIC quality, since the first is more sensitive to physical changes of the media, as shown theoretically and empirically in the paper.



Results show that SMOS SIC have a good correlation and RMS compared with OSI-SAF maps except at thin ice areas. This difference comes from the higher penetration of SMOS, of about 60 cm, with respect to the SIC estimates from higher frequency radiometers; thus, when ice is thinner than 60 cm SMOS data lead to lower values of SIC, what has been verified in this study. These results suggest that by combining SIC information from SMOS and OSI-SAF one could potentially develop

5  a mask for locations of thin ice.

The study presented here can be expanded in various ways, which we are currently exploring. For example, one could conceivably improve the quality of SIC maps by using more tie-points and better characterizing them over different spatial regions and for various times of year. One could also attempt to simultaneously and rigorously estimate SIC and ice thickness (e.g. Rothrock et al., 1988) over thin-ice regions by combining different SMOS observations, thus providing independent

10  estimates of ice volume over these regions. The study could also be further developed in the time-space domain since this has focused in a small fraction of the SMOS dataset making use of just a single point viewed at multiple incidence angles and polarizations, when more than 100 acquisitions can be obtained at each overpass.

*Acknowledgements.* This study has been funded by the National R+D Program of the Spanish Ministry of Economy through the Promises project ESP2015-67549-C3, as well as by previous SMOS-related awards.





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





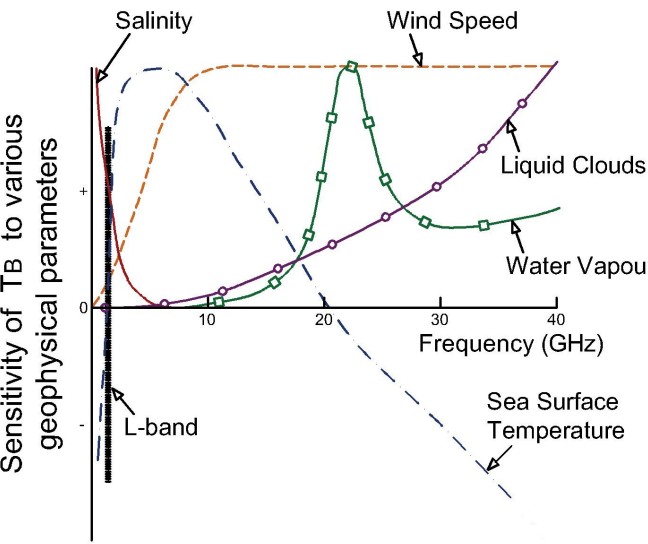

**Figure 1.** Sensitivity of brightness temperature over a range of observing frequencies in the microwave band for a set of key geophysical parameters (created after Wilheit (1978) and Ulaby and Long (2014)). The maximum sensitivity of $T_B$ to sea surface temperature is around 6 GHz, with a peak of 0.4 $K/°C$; to salinity is around 1 GHz, with a peak value of 0.5 K/psu; to wind speed is above 10 GHz, with a peak value of 1 K/m/s. The peak of attenuation from water vapor in clouds is at 22 GHz, and is 0.2 DB/km. L-band (1.4 GHz) observations are hardly sensitive to any variable but salinity, hence it is in a sweet spot for sea ice studies.

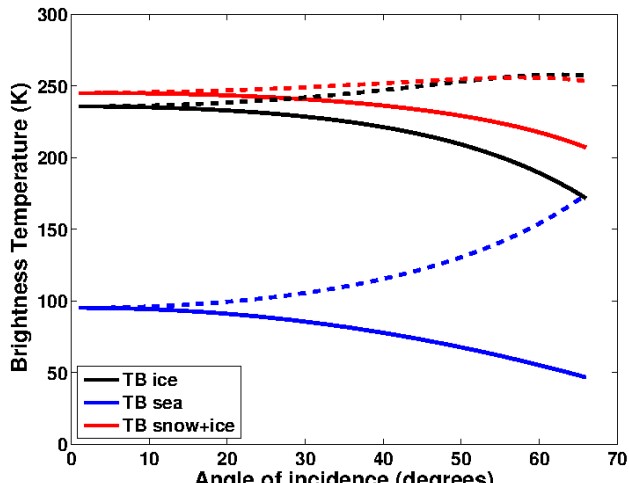

**Figure 2.** Theoretical variation of brightness temperature with angle of incidence for (blue) seawater, (black) sea ice, and (red) a snow layer overlying a sea ice layer for (continuous) horizontal and (dashed) vertical polarizations. (See text in Sect. 3 for details.)




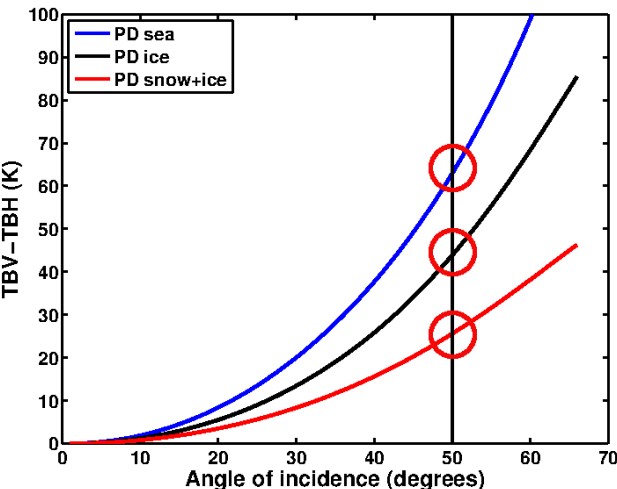

**Figure 3.** Modeled variation of polarization difference ($PD$) index with angle of incidence for (blue) seawater, (black) sea ice, and (red) a snow layer overlying a sea ice layer. (See Eq. 7 and text in Sec. 4.1 for details.) The vertical line at $50°$ incidence angle is drawn for reference to tie points, which are marked with a solid circle for the three media (see text in Sec. 4.2).

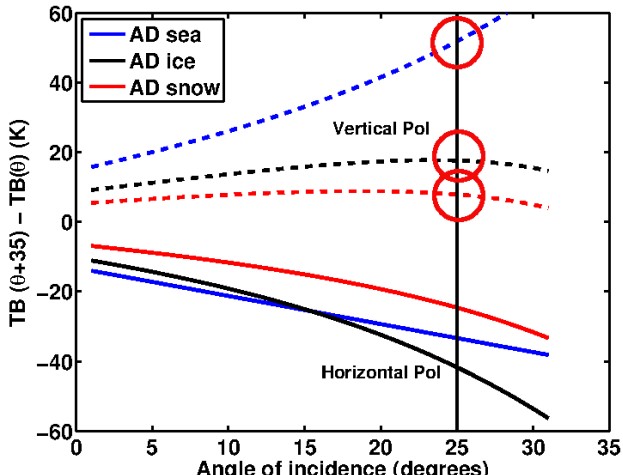

**Figure 4.** Modeled variation of angular difference index ($AD$) with angle of incidence for (blue) seawater, (gray) sea ice, and (red) a snow layer overlying a sea ice layer for (continuous) horizontal and (dashed) vertical polarizations, and for $\Delta\theta = 35°$. (See Eq. 8 and text in Sec. 4.1 for details.) The vertical line at $25°$ incidence angle is drawn for reference to tie points, which are marked with a solid circle on vertical polarization for the three media (see text in Sec. 4.2).



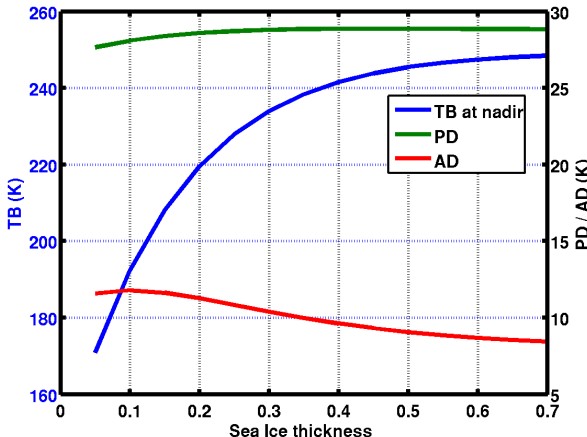

**Figure 5.** Theoretical variation with sea ice thickness of (blue; left axis) $T_B$ at nadir, (green; right axis) polarization difference ($PD$) at $50°$ incidence angle, and (red; right axis) angular difference ($AD$) at $\Delta\theta = 25°$ after the model by Burke et al. (1979), for a sea ice salinity of 8 psu, sea ice temperature of $-10°$ C, and a snow layer of 10-cm thick over the ice. (See text in Sect. 3 for details.) Note the factor of 10 change between the left/right vertical scales.

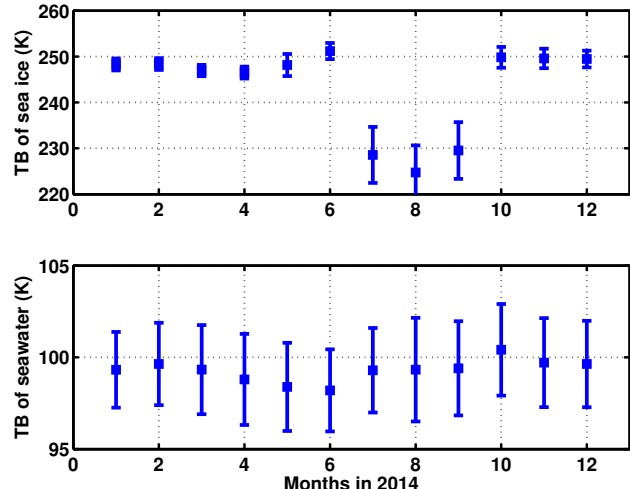

**Figure 6.** Temporal variation of the average brightness temperature $T_B$ at nadir for (top) sea ice and (bottom) seawater at the two tie-point regions (see Sec. 4.4). Note the factor of 4 change in the vertical scales.





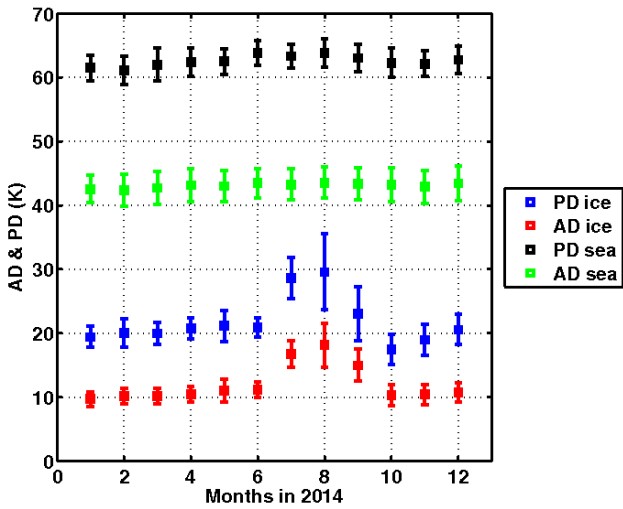

**Figure 7.** Same as Fig. 6 except here for angular and polarization difference indices.

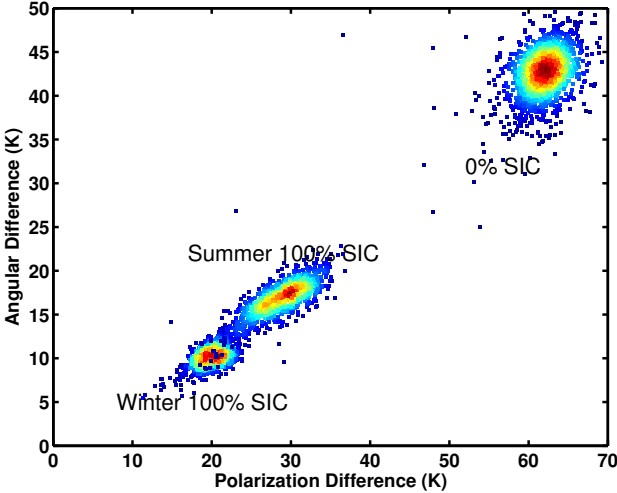

**Figure 8.** Scatter plot of the angular difference vs polarization difference in March and July 2014, with (red-to-blue) high-to-low index occurrence values for the two tie-point regions, i.e., 0% and 100% sea ice concentration (SIC).





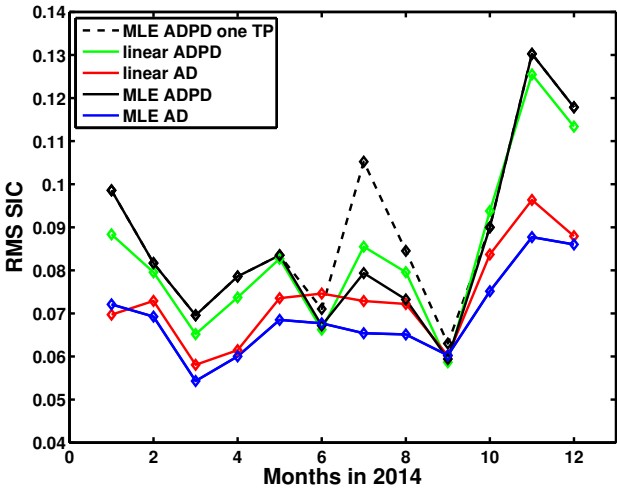

**Figure 9.** Comparison of one tie-point (black dotted line) vs two tie-points (black plane line) with MLE; and MLE vs linear retrieval tecniques. If not defined in the labels is tie-points.

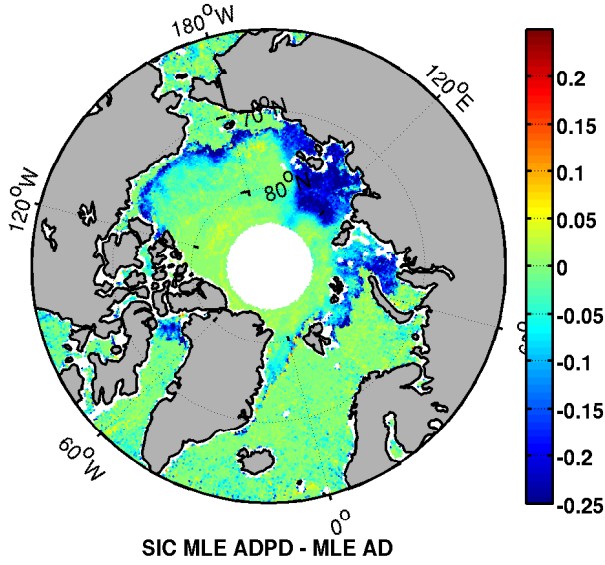

**Figure 10.** SMOS SIC with MLE AD+PD minus SMOS SIC with MLE AD inversion tecniques.





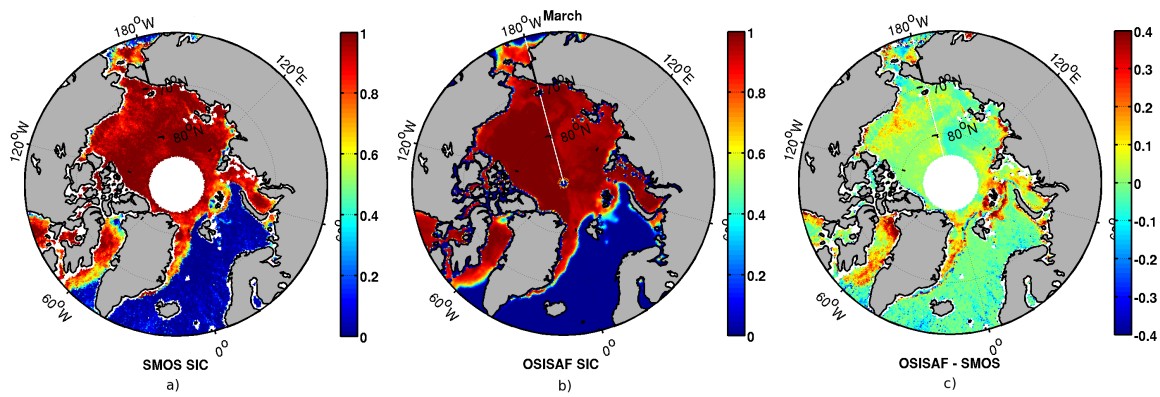

**Figure 11.** SMOS SIC with MLE (a), OSISAF SIC (b) and the differences (c) for 3th March 2014.

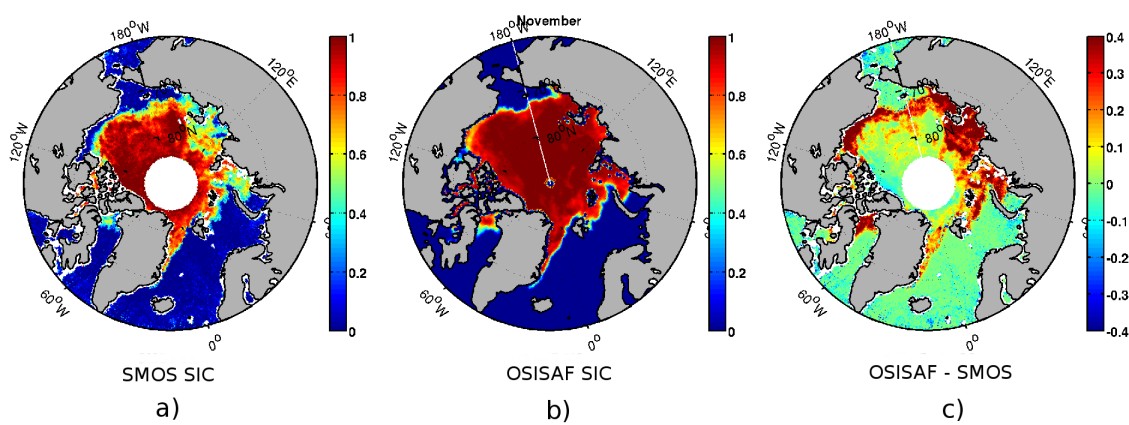

**Figure 12.** SMOS SIC with MLE (a), OSISAF SIC (b) and the differences (c) for 3th November 2014.





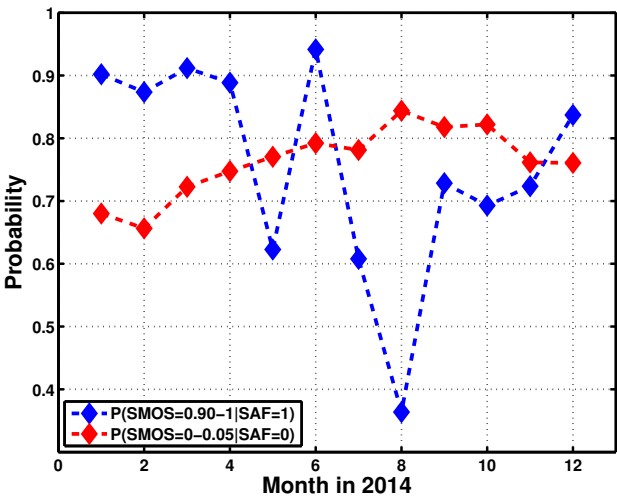

**Figure 13.** Probability to have SMOS SIC more than 0.90 where OSISAF SIC=1 (blue line) and SMOS SIC less than 0.05 where OSISAF SIC=0 (red line) for 2014.

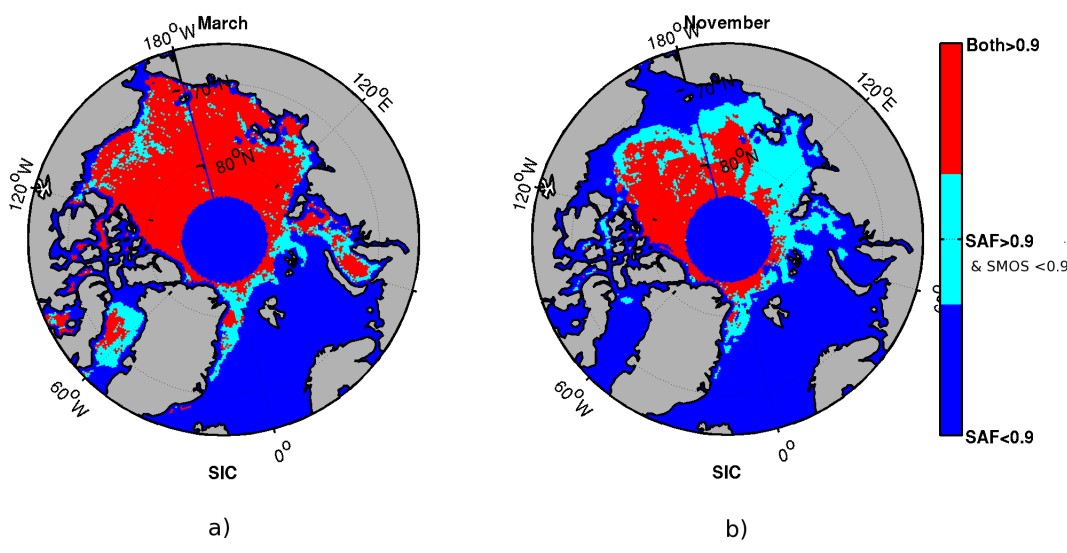

**Figure 14.** Classification of the Artic region according to their values of SMOS and OSI-SAF SIC during March (a) and November (b) 2014. Three classes are shown: 1) OSISAF $SIC < 0.9$; 2) OSISAF $SIC > 0.9$ and SMOS $SIC < 0.9$; and 3) OSISAF $SIC > 0.9$ and SMOS $SIC > 0.9$.





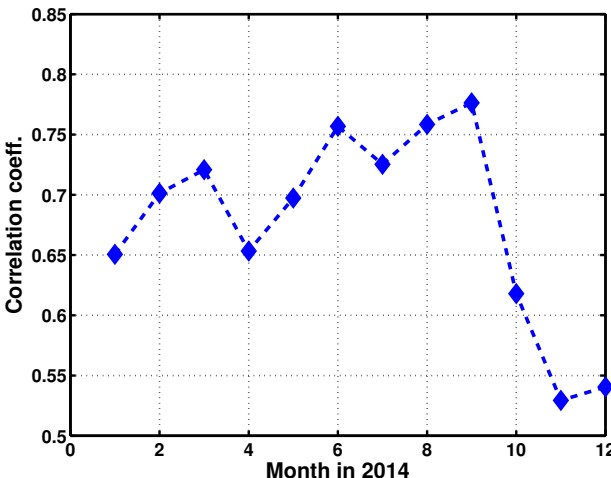

**Figure 15.** Determination coefficient between SMOS and OSISAF SIC for 2014, considering only SIC data in the range from 5% to 95%.

**Table 1.** Modeled (with and without snow) and SMOS observed $T_B$, PD, and AD median values. Errors quoted are the standard deviation around the median.

| | | Modeled (K) | | Observed all year median $\pm\sigma$ (K) | |
|---|---|---|---|---|---|
| 0% SIC | $T_B$ | 95.2 | | 99.33 ± 2.40 | |
| (Seawater) | PD | 62.9 | | 62.56 ± 2.56 | |
| | AD | 51.8 | | 43.08 ± 2.57 | |
| | | Modeled with snow (K) | Modeled without snow (K) | Observed Winter median $\pm\sigma$ (K) | Observed Summer median $\pm\sigma$ (K) |
| 100% SIC | $T_B$ | 249.2 | 239.3 | 248.21 ± 1.56 | 229.04 ± 4.99 |
| (Sea Ice) | PD | 26.8 | 45.9 | 20.30 ± 1.75 | 25.53 ± 3.72 |
| | AD | 8.6 | 18.8 | 10.38 ± 1.17 | 15.26 ± 2.31 |





**Table 2.** Sensitivity of measurement $T_B$, PD, and AD to ice temperature ($T$), salinity ($S$), and thickness ($d$).

| Medium | Index | $\delta I/\delta T$ | $\delta I/\delta S$ | $\delta I/\delta d$ |
|---|---|---|---|---|
| | ($I$) | (K /° C) | (K / psu)[1] | (K / cm) |
| | $T_B$ | 0.2 | 0.51 | |
| Seawater | PD | 0.26 | 0.21 | |
| | AD | 0.20 | 0.12 | |
| | $T_B$ | 0.85 | 1.00 | 1.2 |
| Sea ice | PD | 0.66 | 0.35 | 0.02 |
| | AD | 0.35 | 0.25 | 0.05 |

practical salinity units

**Table 3.** Propagated SIC error using each index, computed from Eq. 9 for assumed ($T$, $S$, $d$) variations, and root-sum-squared (RSS).

| SIC error | index | $\Delta T$ | $\Delta S$ | $\Delta d$ | RSS |
|---|---|---|---|---|---|
| (%) | used | 5 K | 4 psu | 30 cm | |
| $\Delta$SIC | TB | 2.8 | 2.6 | 23.4 | 23.7 |
| $\Delta$SIC | PD | 7.6 | 3.2 | 1.4 | 8.3 |
| $\Delta$SIC | AD | 4.8 | 2.8 | 4.2 | 7.0 |