# Peer review of "Measuring sea ice concentration in the Arctic Ocean using SMOS"

_The Cryosphere, 2016_

## Referee Comment (RC1) · Anonymous Referee #1 · 5 Dec 2016

The use of L-band data for deriving SIC especially during summer, is indeed very welcome and relevant. During summer traditional sea ice concentration (SIC) algorithms using higher frequency channels (19-90GHz) have high uncertainties because the higher frequency brightness temperatures (Tb's) are affected by emissivity variability in the snow/sea ice surface fraction and because of higher levels of water vapor and cloud liquid water in the atmosphere than in winter. L-band Tb's are less sensitive to both noise sources (than 19-90GHz Tb's). In addition SMOS is measuring at several incidence angles at every point which can be exploited in the SIC retrieval (as it was done in this study). So the idea is good, however, this MS is a collection of elements and sections which are not well integrated and focused towards the actual aim of retrieving SIC: for example, the sea ice forward model is used for selecting the parameters AD and PD to use for SIC retrieval and for estimating uncertainties,

however, this may have been more convincing using measurements, or at least a combination of simulations and measurements. This could also give credibility to the model simulations. Several assumptions needs supporting references or investigations (using data), for example, the postulate that Sic is a linear function of AD and PD and that the summer sea ice tie-point is derived from data which are at 100% SIC. Specific comments: P1, L19-20: SIC as an essential climate variable (ECV) has perhaps the longest continuous time-series of satellite measurements among all ECV's and its decline is measured in detail. I don't understand what is meant by lack of observations. You need to specify that. What is meant by quasi-continuous? Please reformulate or delete the sentence. P2, L22: It is important to state here that fig. 1 is for open water (it is not mentioned!). Fig. 1 could be relevant for discussion of the open water tie-point. Please include it in the discussion or delete it. The open water (L-band) emissivity is in addition to salinity also sensitive to SST and wind-speed. P2,L25-26: Please mention the range of frequencies so that this is clear in the sentence. Even though the MY and FY ice Tb's at L-band are overlapping there may be some differences in the mean value. This is a problem for the SIC retrieval since you may be introducing an ice type SIC bias. It needs to be shown, using measurements, how you handle this. P2,L31: delete "quasi" P2,L35: delete "theoretical" P3,L20: Is the galactic reflection correction applied or not? P3L23: It is unclear what is corrected at the bottom of the atmosphere (surface?) using what? P3L26: What is the full range of incidence angles? Are they also extrapolated? P3L30: Add "for comparison" at the end of the sentence. P4L2: Add "dynamic" after "monthly". P4L19: replace "observing" with "electromagnetic" P4,eq.1: Eq. 1 is describing the self-emission of a homogeneous and isothermal material and there is no term for atmospheric reflection or emission/scattering. Perhaps less important terms at L-band but worth mentioning. P4L23: Derbye, 1929, maybe there is a more sea ice relevant reference? P4L23: Add: "…depend on the incidence angle and…" P4L24: replace "complex value" with "complex number" P5L18: "decrease" or "increase"? please give reference. P8L18: reference for the physical parameters variability needs to be given. Only these three

parameters contribute to the budget? Could perhaps also mention snow cover, sea ice type... P9L29: SIC as a linear function of AD and PD. You need to show that this is true (using measurements), otherwise you will have to build in the non-linearities into the SIC model. P11L3-4: why "less prone to errors"? and what is "natural way"? P13L31: Sensitivity to physical temperature. This might be true for some algorithms but not all, please give a reference. P14L1: "sensitive" -> "sensitivity" P14L11: The advantage.. this sentence is nonsense. All tie-points are derived empirically and static tie-points are prone to errors due to sensor drift or seasonal variability, geophysical and climatic trends (in the noise). P14L30: (less noisy) than what? P15L1: what is meant by "good" correlation? Please quantify. P15L13: The AD and PD ice- water contrast is not high for all incidence angles. The Tb contrast is high at all incidence angles. The dynamic range (between ice water) using Tb is much higher than AD and PD and therefor they could be less noisy. Please explain.

Please also note the supplement to this comment:
http://www.the-cryosphere-discuss.net/tc-2016-175/tc-2016-175-RC1-supplement.pdf

---

## Referee Comment (RC2) · Anonymous Referee #2 · 8 Dec 2016

Observation of sea ice concentrations is a highly relevant topic. The support of currently used sea ice concentration retrieval algorithms from other passive microwave sensors operating at higher frequencies by using L-band observations from SMOS is much appreciated. Especially during summer conditions where common products yield higher uncertainties a low frequency algorithm is very welcome. The Manuscript is mostly well written and tries to combines theoretical and empirical aspects to derive sea ice concentration from multi angular observations from SMOS. A solid statistical analysis of the estimated result is given and compared to a sophisticated operational sea ice concentration product. This manuscript is suitable for publication in The Cryosphere after addressing the following comments.

General comments:

[Figure]

1. You employ a physical emissivity model where you find AD and PD relevant but TB too much affected by thin ice. AD and PD are not much affected in the physical model but in the end in the SMOS data the thin ice degrades your retrieved sea ice concentrations a lot. One could argue that your emission model is not able to describe the observations adequately. At this point the question arises if TB would not be even a better indicator for sea ice concentration. It would fortify your approach using the angular difference if you compare the retrieval to a simple TB based approach with the same tie points to show that there is additional information on the ice concentration in the AD compared to TB.

2. During the course of the paper you use different concepts of describing microwave emission which lead to confusion. Firstly you start with emissivity in Eq. 1 and introduce it as 1-reflectivity where the reflectivity is defined for each layer transition while the emissivity should characterize the overall emission. You also use the term "signatures" somewhere in addition to describe MW emission. This could need some clarification.

3. When using the angular difference, you connect data with quite different footprint sizes maybe about 25km vs 60km because of the 35 degree incidence angle difference. I guess this can influence your product at the ice edge and anywhere where you have mixed surface types and should be somehow discussed.

Specific:

P1, L3: remove "interferometric"

P1, L19-21: there are plenty of observations and algorithms observing sea ice and sea ice decline, you cite some of those dataset. Thus this statement is confusing.

P1, L24-25: all the listed publications are sea ice related, thus I would add an important one: Tian-Kunze, X., Kaleschke, L., Maaß, N., Mäkynen, M., Serra, N., Drusch, M., and Krumpen, T.: SMOS-derived thin sea ice thickness: algorithm baseline, product specifications and initial verification, The Cryosphere, 8, 997-1018, doi:10.5194/tc-8-

997-2014, 2014.

P1, L26: add instrument name MIRAS

P2, L5: extension–>extent

P2, L17: add "frequencies" before .

P3, L20: specify which outliers are filtered out, where are they coming from?

P3, L22-23: define "bottom of the atmosphere" and your applied correction for that

P3, L26-27: you write you interpolate TB to locations using a polynomial fit. It is not clear to me if this is a spatial operation or a point wise operation interpolating missing incidence angle ranges in the TB-incidence angle-space.

P4, L6-7: it is not clear for what the NIC data is used

P4, L27-20: The sentence is quite confusing; You say the "latter" which, if I read it correctly, means the dielectric constant is dependent on the incidence angle and thus becomes a tensor. Or do you mean the reflectivity changes with incidence angle (like described by the Fresnel Equations)?

P4, L32: define "standard Arctic temperatures and salinity values"

P5, L3: sensitivity–>variation?

P5, L3: it is unclear for what the reference is there

P5, L8-10: This is a bit confusing, why do you need a constant thickness of the snow layer in an incoherent model (Eq. 2) when the absorption in the snow is negligibly small? Also the mentioning of kappa_e and SSA is confusing here.

P5, L11: remove "spontaneous"

P5, L11-13: Actually the water under sea ice has a contribution to the emissivity, as you can easily calculate with your model, but you mean probably that the emissivity is not

Interactive
comment

getting higher with increasing ice thickness from about 60cm, i.e., the signal saturates. I would rephrase the sentence.

P5, L17: I cannot find anything related to your sentence in the reference you are giving here.

P5, L23-24 (Eq. 2): I cannot see how infinite layer reflections are accounted for. Also that the physical snow temperature times 1-reflectivity of snow-air boundary is simply added is unphysical and must be an error in the equation.

P5, L31-32: remove "conducting". For sure it is also true for a conducting medium but you stated the alpha for low-loss-medium, means no- to low- conducting material.

p6, L1: The dry snow permittivity is actually known to be density dependent, see for example: C. Matzler. Microwave permittivity of dry snow. IEEE Transactions on Geoscience and Remote Sensing, 34(2):573–581, Mar. 1996. ISSN 01962892. doi: 10.1109/36.485133. and M. Tiuri, A. Sihvola, E. Nyfors, and M. Hallikainen. The complex dielectric constant of snow at microwave frequencies. IEEE Journal of Oceanic Engineering, 9(5):377–382, Dec. 1984. ISSN 0364-9059. doi: 10.1109/JOE.1984.1145645.

Eq. 4, Eq. 5, and Eq. 6: I would give the coefficients or skip the equations.

P6, L17: remove "model value necessary for the" or rephrase

P6, L20-23: I don't understand the sentence. The water under the sea ice does not decrease the emissivity of ice but has a fundamental contribution to the emissivity (See also comment on P5, L11-13). I see in Fig. 5 only that emissivity of sea ice increases with ice thickness but not that water under the ice decrease the emissivity of ice. Also: The four layer model does not come with an equation as Eq. 2 only describes ice, snow and air.

P7, L2-3: sentence is confusing, please elaborate or clarify.
P7, L6 (Eq. 7) you should indicate the incidence angle dependence of PD, TBh and TBv

P7, L24: "and snow"–>"with snow cover"

P8, L1-2: add "as described by our model"

P8, L7: remove ", which are rarely available,"

P8, L9: theoretical–>"modeled"

P8, L9-10. The partial derivatives will strongly depend on where they are evaluated as the quantities are nonlinear. This should be mentioned or accounted for. Also the dynamic range of the measurement needs some more explanation.

P8, L21-25: the discussion would need the inclusion of the evaluation point of the partial derivatives

P8, L29: remove "unambiguously", these retrievals also have an uncertainty.

P9, L4: "radiometric values"–>"brightness temperatures"?

P9, L13: add "which" behind first comma

P9, L13: "suggest"–>"suggests"

P9, L14: "maps"–>"retrieval"

P9, L23: remove "algorithm" or rephrase

P11, L11: "extension"–>"extent", "maximum"–>"close to its annual maximum"

P11, L13-14: Thin ice time period was not used for the tie point? this comes as a surprise since your emission model suggested that your key parameters/indices are not sensitive to ice thickness. From where is it known that thin ice introduce a bias in your SIC retrieval, reference?

P12, L3-7: you should mention that "theoretical" means "modeled using Eq. 2"

P12, L25: "adn" –> "and"

P12, L25-26: reference for penetration of frequencies used by OSI-SAF

P12, L28: why are TBs important if your retrieval uses AD?

P13, L19: the referenced figure F. 15 says "correlation coefficient" on the y-axis, so what is really shown?

P14, L3: I could not find this statement in Section 3, SIC is not discussed in Section 3

P14, L16: "of"–>"for"

P14, L18: "changes in the physical media" –> "exchange of the physical medium" or be more specific and write directly about open water and sea ice

P15, L11: I don't understand the sentence: what is meant by "single point viewed"

Fig. 13: would be easier to interpret if the time period with summer tie points is marked or at least mentioned in the caption

---

## Author Comment (AC1) · 6 Feb 2017

Authors: We would like to thank both referees for the interesting and useful questions and improvements suggested.

ANONYMOUS REFEREE 1 The use of L-band data for deriving SIC especially during summer, is indeed very welcome and relevant.

During summer traditional sea ice concentration (SIC) algorithms using higher frequency channels (19- 90GHz) have high uncertainties because the higher frequency brightness temperatures (Tb's) are affected by emissivity variability in the snow/sea ice surface fraction and because of higher levels of water vapor and cloud liquid water in the atmosphere than in winter. L-band Tb's are less sensitive to both noise sources (than 19-90GHz Tb's). In addition SMOS is measuring at several incidence angles at

every point which can be exploited in the SIC retrieval (as it was done in this study).

So the idea is good, however, this MS is a collection of elements and sections which are not well integrated and focused towards the actual aim of retrieving SIC: for example, the sea ice forward model is used for selecting the parameters AD and PD to use for SIC retrieval and for estimating uncertainties, however, this may have been more convincing using measurements, or at least a combination of simulations and measurements. This could also give credibility to the model simulations.

AUTHORS: We have done our best to improve the manuscript along the lines proposed by the reviewer. First a theoretical analysis has been done based on forward models (section 3 and 4.3). On the other hand, in section 4.4 we have computed the parameters based on SMOS measurements. Finally in section 5, we conclude that based on the performance of the different algorithms, it is better to use only the AD index than both AD and PD. Therefore we combine theoretical considerations based on models and measurements in the manuscript. However, we cannot compute uncertainties with SMOS measurements, since the database which is used to validate the SIC data (called RRDB, p. e. Ivanova, et al., 2015) is previous to the SMOS launch.

Several assumptions needs supporting references or investigations (using data), for example, the postulate that Sic is a linear function of AD and PD and that the summer sea ice tie-point is derived from data which are at 100% SIC.

AUTHORS: The referee is right, and that part has now been correctly framed: we do not intend to use a linear model, but to estimate the average slope. Reviewer is right remarking that the computed summer sea ice tie-point probably does not correspond to 100% ice. OSI-SAF SIC data and National Ice Center ice Charts information are used to determine the region with 100% ice. However, OSI-SAF SIC presents large errors during summer (Ivanova et al. 2015), and this could produce errors on the computation of the SMOS 100% SMOS tie point. This is a clear limitation of the method; a short discussion has been added to the text.
Specific comments:

P1, L19-20: SIC as an essential climate variable (ECV) has perhaps the longest continuous time-series of satellite measurements among all ECV's and its decline is measured in detail. I don't understand what is meant by lack of observations. You need to specify that. What is meant by quasi-continuous? Please reformulate or delete the sentence.

AUTHORS: Misleading and wrong sentence, we have deleted it.

P2, L22: It is important to state here that fig. 1 is for open water (it is not mentioned!). Fig. 1 could be relevant for discussion of the open water tie-point. Please include it in the discussion or delete it. The open water (L-band) emissivity is in addition to salinity also sensitive to SST and wind-speed.

AUTHORS: Agreed, it has been specified in the text.

P2,L25-26: Please mention the range of frequencies so that this is clear in the sentence. Even though the MY and FY ice Tb's at L-band are overlapping there may be some differences in the mean value. This is a problem for the SIC retrieval since you may be introducing an ice type SIC bias. It needs to be shown, using measurements, how you handle this.

AUTHORS: The frequency range has been added now in the manuscript, as suggested. Regarding the potentially different radiometric behaviour of MY and FY, we have verified that the difference in the mean values is very small, of around 0.02% (obtained from measurements). Certainly the standard deviations of the two ice types are different: the STD of FY is the double of that of MY ice. But since the mean value is almost the same for MY and FY we do not expect any ice type SIC bias; what should be expected is an increase in uncertainty when FY is dominant, that is, that the error in SIC estimate is larger in that case.

P2,L31: delete "quasi" -> AUTHORS: Done

P2,L35: delete "theoretical" -> AUTHORS: Done

P3,L20: Is the galactic reflection correction applied or not?

AUTHORS: No, the galactic reflection is not significant at high latitudes (as explained paragraph 2 from section 2.1), and is not corrected.

P3L23: It is unclear what is corrected at the bottom of the atmosphere (surface?) using what?

AUTHORS: The geomagnetic and ionospheric rotation and the atmospheric attenuation are corrected to get the bottom of the atmosphere TB. Moreover, some points with low accuracy have been eliminated, such as aliasing (Camps et al., 2005), Sun reflections, and Sun tails. Now the last sentence of the paragraph is deleted, which we think was quite confusing.

P3L26: What is the full range of incidence angles? Are they also extrapolated?

AUTHORS: The full range of incidence angles is from $0°$ to $65°$ (written at the end of section 1). We do not extrapolate data.

P3L30: Add "for comparison" at the end of the sentence. -> AUTHORS: Done

P4L2: Add "dynamic" after "monthly". -> AUTHORS: Done

P4L19: replace "observing" with "electromagnetic" -> AUTHORS: Done.

P4,eq.1: Eq. 1 is describing the self-emission of a homogeneous and isothermal material and there is no term for atmospheric reflection or emission/scattering. Perhaps less important terms at L-band but worth mentioning.

AUTHORS: We have added a new equation 1 in the manuscript which describes the effect of the atmosphere on the final TB, as suggested.

P4L23: Debye, 1929, maybe there is a more sea ice relevant reference? Deleted, the reference had not sense here.

AUTHORS: This reference had not sense here and has been removed.

P4L23: Add: "...depend on the incidence angle and..."-> AUTHORS: Done

P4L24: replace "complex value" with "complex number" -> AUTHORS: Done

P5L18: "decrease" or "increase"? please give reference. -> AUTHORS: Right, it should be 'increase'. Changed in the manuscript.

P8L18: reference for the physical parameters variability needs to be given. Only these three parameters contribute to the budget? Could perhaps also mention snow cover, sea ice type...

AUTHORS: Theses values are just the typical ones, defining the range of variability for them. We do not refer to other parameters (snow cover, sea ice type) since temperature, salinity and ice depth (d) are the only parameters that play a role in our emissivity model. This is now explained in more detail in the paper.

P9L29: SIC as a linear function of AD and PD. You need to show that this is true (using measurements), otherwise you will have to build in the non-linearities into the SIC model.

AUTHORS: Corrected, as commented above.

P11L3-4: why "less prone to errors"? and what is "natural way"?

AUTHORS: The sentence has been deleted. In section 4.5 the characteristics of the MLE technique is described, so it is not necessary to say anything else here.

P13L31: Sensitivity to physical temperature. This might be true for some algorithms but not all, please give a reference.

AUTHORS: We have referred to Ivanova et al. 2015 in this context, and explained in the text now which bands and algorithms suffer of this problem.

P14L1: "sensitive" -> "sensitivity" -> AUTHORS: Done

P14L11: The advantage.. this sentence is nonsense. All tie-points are derived empirically and static tiepoints are prone to errors due to sensor drift or seasonal variability, geophysical and climatic trends (in the noise).

AUTHORS: This sentence has been deleted.

P14L30: (less noisy) than what?

AUTHORS: Than the linear inversion. This sentence has been clarified.

P15L1: what is meant by "good" correlation? Please quantify.

AUTHORS: Sentence has been improved and values are now given

P15L13: The AD and PD ice- water contrast is not high for all incidence angles. The Tb contrast is high at all incidence angles. The dynamic range (between ice water) using Tb is much higher than AD and PD and therefor they could be less noisy. Please explain.

AUTHORS: Although the dynamic range of TB (when comparing ice and water) is larger than those from AD and PD, AD and PD are less sensitive to other parameters (and so less affected by their natural variability), i.e. the ice thickness, the physical temperature and salinity of the Ice/water (refer table 2 and 3). This means that the capability of TB to discriminate between different states is smaller, due to the greater dispersion of the geophysical response, than that of AD and PD, what makes the later better suited to derive SIC.

Please also note the supplement to this comment:
http://www.the-cryosphere-discuss.net/tc-2016-175/tc-2016-175-AC1-supplement.pdf

**Supplement:**

[revised manuscript text omitted]

---

## Author Comment (AC2) · 6 Feb 2017

Authors: We would like to thank both referees for the interesting and useful questions and improvements suggested.

Anonymous Referee #2

Observation of sea ice concentrations is a highly relevant topic. The support of currently used sea ice concentration retrieval algorithms from other passive microwave sensors operating at higher frequencies by using L-band observations from SMOS is much appreciated. Especially during summer conditions where common products yield higher uncertainties a low frequency algorithm is very welcome. The Manuscript is mostly well written and tries to combines theoretical and empirical aspects to derive sea ice concentration from multi angular observations from SMOS. A solid statistical analysis of the estimated result is given and compared to a sophisticated operational sea ice concentration product. This manuscript is suitable for publication in The Cryosphere after addressing the following comments.

General comments:

1. You employ a physical emissivity model where you find AD and PD relevant but TB too much affected by thin ice. AD and PD are not much affected in the physical model but in the end in the SMOS data the thin ice degrades your retrieved sea ice concentrations a lot. One could argue that your emission model is not able to describe the observations adequately. At this point the question arises if TB would not be even a better indicator for sea ice concentration. It would fortify your approach using the angular difference if you compare the retrieval to a simple TB based approach with the same tie points to show that there is additional information on the ice concentration in the AD compared to TB.

AUTHORS: The theoretical models predict that AD and PD are preferable to TB in order to retrieve SIC, not because the larger sensibility of TB to thin ice but also to the other geophysical parameters (temperature, salinity). This does not mean that AD and PD are unaffected by ice depth, as later confirmed in the experiments, but they are still more robust than plain TB (see table 3). The problem with thin ice needs to be attacked, indeed, and for that goal a multiparametric retrieval (both SIC and ice thickness) is in order, as commented in the Conclusions; however, this study goes beyond of the scope of the present paper, and will be addressed in a future work.

2. During the course of the paper you use different concepts of describing microwave emission which lead to confusion. Firstly you start with emissivity in Eq. 1 and introduce it as 1-reflectivity where the reflectivity is defined for each layer transition while the emissivity should characterize the overall emission. You also use the term "signatures" somewhere in addition to describe MW emission. This could need some clarification.

AUTHORS: Agreed. The sentence has been rephrased. We have specified that e

is 1-reflectivity for a unique layer. Latter, we explain how to compute the brightness temperature (not emissivity, which has been changed) for a two-layer model. The word 'signature' has been replaced by 'emissivity' or 'emission'.

3. When using the angular difference, you connect data with quite different footprint sizes maybe about 25km vs 60km because of the 35 degree incidence angle difference. I guess this can influence your product at the ice edge and anywhere where you have mixed surface types and should be somehow discussed.

AUTHORS: The reviewer is completely right: At 25° incidence angle SMOS resolution is around 38 km and at 60° is around 70 km, or an increase of 84%. So certainly the measurements do not refer to the same area, and this is why probably the use of AD is better suited for cases in the interior areas and is more problematic close to the coast. A comment on this issue has been added in the text.

Specific:

P1, L3: remove "interferometric"

AUTHORS: Done.

P1, L19-21: there are plenty of observations and algorithms observing sea ice and sea ice decline, you cite some of those dataset. Thus this statement is confusing.

AUTHORS: Agree. The last two sentences have been deleted.

P1, L24-25: all the listed publications are sea ice related, thus I would add an important one: Tian-Kunze, X., Kaleschke, L., Maaß, N., Mäkynen, M., Serra, N., Drusch, M., and Krumpen, T.: SMOS-derived thin sea ice thickness: algorithm baseline, product specifications and initial verification, The Cryosphere, 8, 997-1018, doi:10.5194/tc-8-997-2014, 2014.

AUTHORS: Now included.

P1, L26: add instrument name MIRAS AUTHORS: Done in page 2 .

P2, L5: extension–>extent AUTHORS: Done.

P2, L17: add "frequencies" before .

AUTHORS_ Done .

P3, L20: specify which outliers are filtered out, where are they coming from?

AUTHORS: We filter out all the Tb measurements, from each grid point, which are further away than 3*sigma. It is added in P3L23 of the new version.

P3, L22-23: define "bottom of the atmosphere" and your applied correction for that

AUTHORS: This sentence was wrong. The final TB is taken at the reference frame "bottom of the atmosphere", when atmospheric and geomagnetic and ionospheric issues are already corrected for. We have rephrased the sentence.

P3, L26-27: you write you interpolate TB to locations using a polynomial fit. It is not clear to me if this is a spatial operation or a point wise operation interpolating missing incidence angle ranges in the TB-incidence angle-space.

AUTHORS: We interpolate the SMOS data of the same grid point (pixel) to obtain the TB in the incidence angles which are missing.

P4, L6-7: it is not clear for what the NIC data is used

AUTHORS: It is now explained in line P4L7.

P4, L27-20: The sentence is quite confusing; You say the "latter" which, if I read it correctly, means the dielectric constant is dependent on the incidence angle and thus becomes a tensor. Or do you mean the reflectivity changes with incidence angle (like described by the Fresnel Equations)?

AUTHORS: Agree. The sentence was confusing. We have made an effort to make it clearer.

P4, L32: define "standard Arctic temperatures and salinity values"

AUTHORS: This information is now added in the manuscript.

P5, L3: sensitivity–>variation?

AUTHORS: Done.

P5, L3: it is unclear for what the reference is there

AUTHORS: It was referred to the snow effect on the SMOS TB. Now the text has been modified to make it clearer.

P5, L8-10: This is a bit confusing, why do you need a constant thickness of the snow layer in an incoherent model (Eq. 2) when the absorption in the snow is negligibly small? Also the mentioning of kappa_e and SSA is confusing here.

AUTHORS: Agreed. This part is deleted in the new version. This is true for any media, but has no sense when snow media is in the middle layer.

P5, L11: remove "spontaneous" AUTHORS: Done.

P5, L11-13: Actually the water under sea ice has a contribution to the emissivity, as you can easily calculate with your model, but you mean probably that the emissivity is not getting higher with increasing ice thickness from about 60cm, i.e., the signal saturates. I would rephrase the sentence.

AUTHORS: Done.

P5, L17: I cannot find anything related to your sentence in the reference you are giving here.

AUTHORS: It was a mistake, the reference Maa$\beta$ et al. 2015 has been deleted here.

P5, L23-24 (Eq. 2): I cannot see how infinite layer reflections are accounted for. Also that the physical snow temperature times 1-reflectivity of snow-air boundary is simply added is unphysical and must be an error in the equation.

AUTHORS: The reviewer is right, the equation as written in the paper was wrong, and

the infinite reflections were not taken into account there. We apologize for the mistake; it is now properly written.

P5, L31-32: remove "conducting". For sure it is also true for a conducting medium but you stated the alpha for low-loss-medium, means no- to low- conducting material.

AUTHORS: Done.

p6, L1: The dry snow permittivity is actually known to be density dependent, see for example: C. Matzler. Microwave permittivity of dry snow. IEEE Transactions on Geoscience and Remote Sensing, 34(2):573–581, Mar. 1996. ISSN 01962892. doi: 10.1109/36.485133. and M. Tiuri, A. Sihvola, E. Nyfors, and M. Hallikainen. The complex dielectric constant of snow at microwave frequencies. IEEE Journal of Oceanic Engineering, 9(5):377–382, Dec. 1984. ISSN 0364-9059. doi:10.1109/JOE.1984.1145645.

AUTHORS: Agreed. The paper has been modified accordingly, including the appropriate citations.

Eq. 4, Eq. 5, and Eq. 6: I would give the coefficients or skip the equations.

AUTHORS: The authors prefers to keep the equations on the manuscript, since they shows the dependences to other parameters. However, we prefer not to add the values of the coefficients, since they do not bring any additional information to the readers, and all the values are in the cited papers.

P6, L17: remove "model value necessary for the" or rephrase

AUTHORS: Agreed. Done.

P6, L20-23: I don't understand the sentence. The water under the sea ice does not decrease the emissivity of ice but has a fundamental contribution to the emissivity (See also comment on P5, L11-13). I see in Fig. 5 only that emissivity of sea ice increases with ice thickness but not that water under the ice decrease the emissivity of ice. Also:

The four layer model does not come with an equation as Eq. 2 only describes ice, snow and air.

AUTHORS: The sentence was not correct, actually. The decrease of emissivity is due to the reduction of the ice thickness. This sentence has been corrected.

P7, L2-3: sentence is confusing, please elaborate or clarify.

AUTHORS: Certainly the sentence was confusing. We have deleted it since the information we wanted to transmit here is already given in the same paragraph in: 'It is possible, however, to define a number of indices combination of brightness temperature observations that are less sensitive to the unknown physical parameters. '

P7, L6 (Eq. 7) you should indicate the incidence angle dependence of PD, TBh and TBv

AUTHORS: Agreed. Done.

P7, L24: "and snow"–>"with snow cover" -> Authors: Done.

P8, L1-2: add "as described by our model" -> Authors: Done.

P8, L7: remove ", which are rarely available," -> Authors: Done.

P8, L9: theoretical–>"modeled" -> Authors: Done.

P8, L9-10. The partial derivatives will strongly depend on where they are evaluated as the quantities are nonlinear. This should be mentioned or accounted for. Also the dynamic range of the measurement needs some more explanation.

AUTHORS: The sensitivities, listed in table 2, are obtained using the following range of parameters, for sea water : Tsea=[2,15], Ssea=[10,38] and for ice: Tice=[-20,-5] and Sice=[2,12]. This information is added in page 8.

P8, L21-25: the discussion would need the inclusion of the evaluation point of the partial derivatives

AUTHORS: This is addressed in the previous answer.

P8, L29: remove "unambiguously", these retrievals also have an uncertainty. -> AUTHORS: Completely agreed. Done.

P9, L4: "radiometric values"–>"brightness temperatures"? -> AUTHORS: Done.

P9, L13: add "which" behind first comma ->AUTHORS: Modified

P9, L13: "suggest"–>"suggests" -> AUTHORS: Done.

P9, L14: "maps"–>"retrieval" -> AUTHORS: Done

P9, L23: remove "algorithm" or rephrase - > Authors: Done

P11, L11: "extension"–>"extent", "maximum"–>"close to its annual maximum"-> AUTHORS: Done

P11, L13-14: Thin ice time period was not used for the tie point? this comes as a surprise since your emission model suggested that your key parameters/indices are not sensitive to ice thickness. From where is it known that thin ice introduce a bias in your SIC retrieval, reference?

AUTHORS: The models do not suggest that the indices are NOT sensitive, they suggest that the sensitivity of the indices to thin ice is lower than using TBs. Figure 5 and table 2 show that there is, still, a sensitivity of AD and PD to thin ice, even though this is smaller than TB. The sentence has been modified according.

P12, L3-7: you should mention that "theoretical" means "modeled using Eq. 2" -> AUTHORS: Done

P12, L25: "adn" –> "and" ->AUTHORS: Done.

P12, L25-26: reference for penetration of frequencies used by OSI-SAF -> AUTHORS: Done

P12, L28: why are TBs important if your retrieval uses AD? -> AUTHORS: True, it has

been modified.

P13, L19: the referenced figure F. 15 says "correlation coefficient" on the y-axis, so what is really shown? -> AUTHORS: changed

P14, L3: I could not find this statement in Section 3, SIC is not discussed in Section 3-> AUTHORS: True, we have modified 'SIC' by 'TB'.

P14, L16: "of"–>"for"-> AUTHORS: Done.

P14, L18: "changes in the physical media" –> "exchange of the physical medium" or be more specific and write directly about open water and sea ice -> AUTHORS: Done.

P15, L11: I don't understand the sentence: what is meant by "single point viewed" -> AUTHORS: It has been rewritten now.

Fig. 13: would be easier to interpret if the time period with summer tie points is marked or at least mentioned in the caption. -> AUTHORS: Done.

---

## Referee Report (RR1)

Review for Gabarro et al. : **Measuring sea ice concentration in the Arctic Ocean using SMOS**
Submitted to *The Cryosphere*

**Summary**

The authors develop a new algorithm to retrieve sea ice concentration from the L-Band SMOS measurements at 1.4 GHz. At 1.4 GHz, the influence of the atmospheric properties on brightness temperatures is very low and, additionally, SMOS provides full-polarized measurements at different incident angles. Due to a higher penetration depth, information about the sea-ice thickness can be retrieved in addition to sea-ice concentration. Ideally, the method here is a first step to combining sea-ice concentration (SIC) and sea-ice thickness measurements from the same place at the same time.
For their new SIC retrieval method, the authors take advantage of the special features provided by SMOS and introduce two indices, the polarization difference and the angular difference, to avoid the dependence of the brightness temperature on the sea-ice thickness. They use a Maximum Likelihood Estimator in combination with these two indices in opposite to the more usual method of linear estimation used in algorithms designed for higher frequencies. They find that the retrieved SIC compare well with observations, except in fall, where there are differences in regions of thin ice due to the high penetration depth of the low-frequency radiation.

The topic is timely and the approach of the Maximum Likelihood Estimator is interesting. The authors took well into account previous comments and therefore improved the manuscript notably. However, I think there is still room for improvement in the structure and writing style. The clarity of your message would profit from a careful structure- and writing-oriented (instead of topic-oriented) proof-reading.

I suggest minor revisions. I have some comments and questions and I hope the authors can answer them. Also, I have some suggestions that could improve the clarity of the manuscript.

**Thematic comments**

**#1** It is not totally clear to me what is the advantage of this new method, with which I mean a SIC retrieval at 1.4 GHz. I can understand that there has not been any SIC retrieval at this frequency before but is it then not only one new method amongst others to retrieve SIC? If I understood right, this retrieval yields smaller errors in summer. You could underline a bit more that this is a key advantage compared to other algorithms, which have problems in summer due to melt ponds and wet snow for example.

**#2** On the same note, you state as an advantage that, as we now could in principal retrieve both sea-ice thickness and SIC from 1.4 GHz-measurements, these could be combined to retrieve both at the same time. But can thickness and concentration be retrieved at the same time if the retrieval method for SIC has problems in the thin ice areas (under 60 cm) and the retrieval method for sea-ice thickness is only for thin ice areas, up to 50 cm (Kaleschke et al., 2012; Huntemann el al., 2014)? I would like the authors to comment on that.

**Writing comments**

**#3** The words "below", "above", "former", "latter" are used a lot. Often this is too vague and it is not clear what exactly is meant by them. The reader is pointed in a lot of directions and gets off the track of the actual message. I suggest that you rethink the structure of the manuscript to avoid as much as possible having to point to another place in a manuscript.

**#4** It is sometimes unclear what was done in the study and what has been done before. As you use several tenses (past, perfect, present, future) and active and passive mode in an inconsistent way (sometimes changing in the middle of a paragraph), I suggest to carefully proof-read the manuscript and and to correct the inconsistencies. This would remove some of the confusion.

**#5** Also, very often a paragraph or sentence starts with "the figure shows", "the table shows" or "the authors show". I think if the emphasis was on the message of the figure/reference and the figure/reference was only given in parenthesis at the end of the sentence, your message would gain in clarity.
Consider as an example the difference between:
P13 L1-4 : "Figure 11 shows the spatial distribution of SIC in the Arctic Ocean estimated from (a) SMOS for the 3-day period 2–5 March 2015, (b) OSI-SAF SIC on 4 March 2014, and (c) the difference between (b) and (a). "
P14 L22-23 : "However, the sensitivity of the brightness temperature to sea surface temperature, atmosphere, and wind speed is clearly reduced when observing the sea surface with radiometers working at lower frequencies (Figure 1) ..."

**#6** I find it confusing when parentheses are used for a whole sentence. Either it is important for the study, then the sentence can be written as such, or it is not important and it can be left out.

**#7** Some sentences are very long with several dependencies and often too many or too less commas. I lost track several times. Shorter sentences would improve clarity.

**#8** There are still several typos and grammatical mistakes. I tried to highlight some of them but I suggest that you let a native speaker or just another person read through your manuscript.

**Specific comments**

P1 L11: I don't see the logical connection from the previous sentence to the "therefore"

P1 L18: Replace "ice cover" by "sea-ice cover"

P2 L5-7: The sentence is unclear as you use both "therefore" and "because". Try to divide it into two sentences.

P2 L7 : Replace "since" by "for"

P2 L16 : Comma after "that" and it is not clear to what "former" refers. Add "the ice penetration of the former".

P2 L22: I think you can leave out "what is left for a future work"

P2 L29: Not clear why the spatial resolution of 35 to 50 km is a key feature. Maybe remove "key" in L28.

P3 L10: Remove the sentence in parentheses.

P3 L30-31: Move the product version to the OSI SAF parenthesis.

P4 L3 and L7 : I think you can remove "see below".

P4 L5 : You always only mention data from 2014. Would it be right to only write "from the year 2014"? And then you could leave out in the rest of the manuscript all the time you mention "over the year 2014". If you use more years (that did not become clear to me), write "from 2014 to XXXX". Otherwise it is not clear when your data period ends.

P4 L13 : Remove "As discussed in Section 1"

P4 L13 : Replace "different incidece angle at" by "different incidence angles to"

P4 L14: I think you can start a new sentence: "TB can be expressed as".

P4 L18: Replace "into" by "on"

P4 L21: Write sentence without parentheses. Do you mean "We use TB to refer to surface brightness temperature"? The surface emissivity would be e_s, right? This is not clear.

P5 L4-5: There is no logical sequence between "varies **linearly** with emissivity" and "The **nonlinearity**".

P5 L7: Replace "with" with "on the"

P5 L8-10: Write the sentence without parentheses

P5 L20: I think you can remove the sentence in parentheses.

P5 L30: I think it would be clearer to introduce the equation directly when you cite it for the first time (near L8). Maybe it would work if you change the sequence of the paragraphs on P5 and P6.

P6 L25-29: I think this sentence is too long.

P7 L2: I think "microwave remote sensing model" is not the right term here. Maybe you mean "microwave emission model"?

P7 L10: Remove "deletedwhen"

P7 L9-11: I don't understand this sentence.

P7 L20: It is not clear what "the former" refers to.

P7 L24-29: "In this study","in this context","in this applications". The logical sequence of these sentences is not clear.

P8 L1: you could cite (Tab. 1) after ice

P8 L6: "indicated above" is not specific enough, it could be 25°, 60°, 30°, 50°.

P8 L28: I don't know if "reasonable" is the right word here. Maybe "average"?

P9 L5: "done by other authors". It is not clear to what "done" refers. Did they focus on TB or on inversion algorithms using PD and AD? I am quite sure that it is the former but this is not clear from the sentence structure.

P9 L14: If the level of uncertainty is unquantified, does it still mean that it is negligible?

P9 L27: add "defined" before "above"

P9 L32-P10 L2: The paragraph starts with "Table 1 lists". It reads as if you were introducing something new. But, actually, you have referenced Table 1 several times before. This relates again to the structural issue (see #3).

P11 L5: If possible, try to use another letter for the distributions. rho was already used for the density before.

Section 5.1.: This is more a listing of different figures than a coherent story (see #5). I suggest that you rethink about the message you want to convey in this section.

P12 L1: Remove "replacedmoenths" and replace "epochs" by "periods".

P12 L29: I think you can remove "As we have shown"

P13 L5: I think you can remove "here" and "some days in"

P13 L9: It is not clear to what "that response" relates to.

P13 L31-33: Reformulate this sentence. Not clear.

P14 L6: I think you can replace "that is, the square of correlation coefficients" by "R^2" or "r^2"

P14 L11: You don't need to put parentheses here.

P14 L19: It is not clear if this has been identified by Ivanova et al. (2015) or by someone else.

P14 L30: You introduce the full names of AD and PD only later in the conclusions (P15 L6).

P15 L24: Do you mean "determination" or "correlation" coefficients?

**Figures**

The caption often contains more explanations than are needed. And I think you do not need to reference in which part of the text the figure is discussed. So I think you can remove all "see XXX".

Figure 1: A similar sentence can be found in the text, I think you can remove it.

Figure 4: Replace "gray" by "black".

Figure 10, 11 and 12: I suggest that you use different colorbars. This would improve the clarity of the figures. You could use blue-white-red for difference plots and blues or reds for absolute values.

Figure 11 and 12: Replace "3th" by "3rd"

Figure 14: In the colorbar, replace "SAF<0.9" by "Both<0.9".

**References**

Several references are missing dois. In other cases, the format of the dois is not consistent, e.g. for
Huntemann et al, 2014, Shokr et al., 2015 and many others.
Also sometimes, journal abbreviations are used, sometimes not.
Tonboe et al., 2016 is not a discussion paper anymore.
Ulaby et al., 2014: Replace "adn" by "and"
Vant et al, 1978: A "T" is missing in the beginning of the title.

---

## Referee Report (RR2)

Second review for Gabarro et al. : **Estimating sea ice concentration in the Arctic Ocean using SMOS**
Submitted to *The Cryosphere*

I thank the authors for answering my questions clearly and taking into account my comments about the structure. The manuscript has improved a lot since the last version. The story is much clearer now and reads much more fluently.
I still have some minor comments. Overall, however, the manuscript is nearly ready for publication.

**Thematic comments**

**1 L335: "most of the values are in agreement at about 2*sigma" is very vague (mostly because of "most" and "about"). I would suggest reformulating this more carefully. Maybe you could briefly discuss winter and summer results separately, as in summer they fit less, as expected.**

**2 L380-384: Something is not right here. You mention a likelihood function L and a likelihood function ^l. And you mention "Recall that the likelihood is the logarithm of the probability density function". So, L = ln(N) ? But then Eq. (18) says ^l = ln(L). I suggest checking this again.**

**3 L413-414: I am wondering if you are basing "since this period is the one with the minimum ice extension and minimum thin ice expected" only on this figure or also on other results? I wonder if, with this explanation, we should not also see effects in July and August as well. It does not seem obvious to me that they agree so well in September but that the disagreement is higher in the other summer months. But maybe I understand something wrong. Could you comment on this?**

**4 L463: This is a bit vague. I suggest adding numbers, for example by replacing "is almost zero" by "is between 20 and 30%". Also note that you use "very high" (without mentioning 0.7 to 0.8) and "notably high" (with mentioning 0.9). This might be misleading without numbers.**

**Style/Typos**

L38-39: I think you should add an information about the area of ocean considered in the definition of SIC. "the total area at a given ocean location" is a bit vague. For example : "the fraction of ice relative to the total area of a given ocean domain".

L101: Replace "filtered" with "filter"

L103: Replace "averaged" with "average"

L116: Replace "used" with "use"

L118: Replace "used" with "use"

L125: Replace "transmitivity" by "transmittivity"

L144: Delete "will"

L158: Replace "imagenary" with "imaginary"

L179: Insert "is" in "c is the speed of light"

L184: I do not understand why there is a circumflex accent on eps_ice

L225: I would suggest saying: "Hereafter, we introduce..." so that it is clear, that this is your result

L303: Replace "selected" with "select"

L305: Replace "selected" with "select"

L310: Replace "calculated" with "calculate"

L329: Remove "group"

L377: Replace "used" with "use"

L378: Replace "means" by "stands for"

L435-436: Use "we have estimated" and "we have compared" instead of passive voice to make clear that you did this and it was not done before.

L449: Add "it" in "whereas at the OSI-SAF frequencies (…), it is..."

L455: Replace "seas" by "sea"

L462: Add "also" in "This also implies"

L473: "improveS"

L481: Replace first "is" with "are"

L497: Add "as" in "such as flat surface ice"

All of Section 5, you use perfect tense. I suggest being consistent with other sections and using present.

**Figures**

Fig. 1: I suggest replacing "for a set of" with "to a set of", because it is a sensitivity **to** something.

Fig. 5: I suggest adding the unit to the x-axis label

Fig. 6: Replace "multy" with "multi"

Fig. 9: Replace "tecniques" with "techniques"

Fig. 11-12: I still suggest changing the colorbar to "blue to white" as it is more intuitive.

---

## Author Response (AR2)

28 May 2017

Prof. Lars Kaleschke
Scientific Editor
*The Cryosphere*

**Manuscript tc-2016-175**

Dear Prof. Lars Kaleschke,

Please find attached the revised manuscript entitled *Estimating sea ice concentration in the Arctic Ocean using SMOS.* (Please note the title change.) This manuscript has been revised and improved based on the comments we received from you and the reviewers. These revisions do not change the main scientific results of the research reported in the manuscript.

We also attach the new version of the manuscript with marked changes (in blue color) with respect to previous version. Below we describe the revisions made to the manuscript and the response to the reviewers' comments.

Sincerely,

Dr. Carolina Gabarro
Barcelona Expert Center
Physical Oceanography Department
Marine Sciences Institute (ICM)
National Research Council (CSIC)
E-mail: cgabarro@icm.csic.es

**Response to the Editor (Prof. Lars Kaleschke)**

(*Original comments are shown in italics*; **Authors: responses in boldfaced font; number of lines and pages is from the track changes version or the manuscript.**)

*The thematic comments of referee #4 reflect my main concerns: What is the advantage of the new method?*

**Authors: This comment has been addressed in the response to Reviewer #4. In essence, the main advantage of L-band observations is the significantly less sensitivity to atmospheric effects than observations from radiometers at higher frequencies, and to temperature changes than radiometers between 6-10 GHz. Furthermore, the effect of snow depth, which is an important contributor to the SIC error budget from higher-frequency radiometers, can be assumed to be negligible here. That is because this method relies on the use of angular differences (AD) of only vertical polarization (V-pol) observations and, as shown my Mass et al. (2015), the effect of snow depth is observable at horizontal polarization (H-pol) but not vertical. Moreover, the method presented here is robust because uses a maximum likelihood estimator (MLE) to invert for sea ice concentration (SIC) parameters. This is also a new feature of this method, and could be beneficial for data obtained from other radiometers. All in all, as Reviewer#3 also pointed out, this is a new independent method that can help cross-check SIC results obtained with other methods.**

*How can the interference between thickness and concentration be resolved?*

**Authors: This question has also been addressed in the response to Reviewer #4. We have removed the single comment there was in the original manuscript to a suggestion of attempting the simultaneous estimation of SIC and sea ice thickness (SIT). We realize this is an important but also difficult problem that deserves some serious research. The comment was largely motivational, but has now been dropped.**

*One main parameter in question is the PD which is also used for the retrieval of ice thickness, see Huntemann et al. (2014) equation 4. This usage seems to be in strong contrast to your results shown in Fig. 5 which suggest only a weak dependency of PD on thickness.*

**Authors: The thrust of this study is a new method for SIC not SIT retrievals. Figures 10 and 11, which are derived from observations, suggest that there is some PD sensitivity to ice thickness and that SIC differences are associated with regions of thin ice. Figure 5 estimates are derived from a theoretical model that incorporates various simplifying assumptions and nominal model parameter values. Like with most models, while being useful they also have their limitations. We feel there is no need to discuss methods for**

retrieval of SIT including Huntemann because our study focuses on SIC, not on SIT retrieval.

*The model assumes dry snow but the algorithm is applied during the melting period. An extended discussion on the Summer period may perhaps help to show the advantage of SMOS for ice concentration retrieval. You state that there are no validation data before the launch of SMOS to calculate uncertainties. But you could make comparisons to other independent ice concentration estimates, e.g. using high resolution optical data in cloud free areas.*

**Authors: It should first be clarified that our algorithm for SIC retrieval is fully empirical, no assumptions are made on the type of snow. The assumption of dry snow is only made when discussing theoretical models, which are discarded for the construction of the algorithm.**

**The comment is nevertheless appreciated, and we take good note for future work. In our response to Reviewer #4 on this topic, we indicated that a comparison with optical imagery would help for validation purposes of SMOS SIC during summer periods but that such study is beyond the scope of this manuscript.**

*To improve the reproducibility I place emphasis on the comments of referee #2 who asked for the model equations.*

**Authors: This has been done, as indicated in our response to Reviewer #2.**

*Regarding the structure I recommend to split discussion and conclusions. The title should be changed according to the suggestion of referee #3.*

**Authors: The sections have been split, the title changed, and the manuscript edited and proofread.**

**Response to Reviewer #2 (anonymous)**

**General Comments:**

*Sea ice concentration observations are a highly relevant topic. In summer, sea ice concentration products have typically higher uncertainties compared to winter. Extending the frequency range of passive microwave retrieval algorithms to lower frequencies such as L-band, has the potential to lower the overall uncertainties of sea ice concentrations. The Manuscript is well written and combines theoretical and empirical aspects to derive sea ice concentration from multi angular observations from SMOS. The presented algorithm includes an estimation of uncertainties of sea ice concentration retrievals and is compared a sophisticated operational sea ice concentration product.*

**Authors: We thank the reviewer for the positive comments and summary.**

**Specific Comments:**

*The manuscript is suitable for publication in The Cryosphere after addressing the following specific comments:*

*P2, L6: remove "AMSR-2 and"*

**Authors: Done.**

*P3, L27: how can it happen that the incidence angle range is not fully covered within three days? It should be mentioned in the manuscript if also extrapolation is done or not. "interpolate TB to locations" here is misleading as it sound like interpolating spatially and not in TB-incidence angle space.*

**Authors: The sentence has been modified for improved clarity, in line 112, page 4.**

*P4, L16-17: all "self-emitted"□"emitted"*

**Authors: Done in line 137, page 5. In fact, the "self-" had been added on a suggestion from a previous reviewer.**

*P5, L8-10: remove parenthesis*

**Authors: Done.**

*P5, L12: "models" -> "cases"*

**Authors: Done.**

*P5, L12: remove "larger"*

**Authors: Done.**

*P5, L18: remove "spontaneous"*

**Authors: Done.**

*P5, L11-13: Actually the water under sea ice has a contribution to the emissivity, as you can easily calculate with your model, but you mean probably that the emissivity is not getting higher with increasing ice thickness from about 60cm, i.e., the signal saturates. Please rephrase the sentence. (see also comment on P6, L25)*

**Authors: That is correct, the emissivity does not further increase for ice thicker than about 60 cm because of attenuation. The sentence has been rephrased in line 180, page 6.**

*P5, L23: Snell's law describes the propagation direction inside a medium with respect to its permittivity and does not describe transmissivity/emissivity.*

*Would be good to have either a reference (maybe Schwank et al. (2015)\*) or more explanation.*

**Authors: Sentence rephrased, and reference added, in line 190, page 6.**

*P5, L29-30 (Eq. 3): Burke et al. (1979) actually does not make the step to expand the multiple reflection to the binomial series. So the source of the Equation could be Ulaby (1981)\*\* Eq. 4.163, or derived from Ulaby (2014). A follow up on this equation is also useful as most of the terms disappear or become much simpler once you assume negligible attenuation in the snow. Also the later used water layer is not included in this equation.*

**Authors: The reviewer is correct, it is Ulaby and not Burke who expands to multiple reflections in the case of three media. We have changed the text accordingly, in line 200, page 7.**

*P6, L5-6: remove "conducting". For sure it is also true for a conducting medium but you stated the alpha for low-loss-medium, means no- to low- conducting material. Also the penetration depth is not used further in the document, so the sentence may be removed.*

**Authors: Removed "conducting." About the skin depth, we think it will help the reader the general to better grasp the meaning of the equations, thus kept it.**

*P6, L7: The dry snow dielectric properties are discussed twice, here and in P5, L22-It is probably good to put this together for consistency.*

**Authors: Done. Now the snow discussion is in lines 184-194, page 6.**

*P6, L22: Equation for the four layer model is missing.*

**Authors:** Included, as Eq. (9), in page 7.

*P6, L25: The surface emissivity of seawater is not a relevant quantity for the transmissivity of radiation from seawater under the ice into the ice layer. The boundary between seawater and sea ice has a different reflectivity compared to the boundary between seawater and air because of the different permittivities of ice and air. I suggest to just remove the mentioning of the emissivity of sea water and start with "The net effect ...*

**Authors:** Accepted and implanted, in line 148, page 8.

*P8, L20-23. The variation of a quantity with another within a multi dimensional space depends on all other variables in case they are not dependent nor correlated. However, Table 2 shows only a single value for each quantity. Please give more details on how these values were obtained.*

**Authors:** To compute the sensitivities of the indices (Tb, PD, AD) to several parameters we use the range of values of each parameter, and the rest of the variables are set to the values in page 5 line 9 - 10. It is now specified in the text, line 320, page 10.

References:

*Schwank M, Mätzler C, Wiesmann A, et al. Snow Density and Ground Permittivity Retrieved from L-Band Radiometry: A Synthetic Analysis. IEEE J Sel Top Appl Earth Obs Remote Sens. 2015:1-14. doi:10.1109/JSTARS.2015.2422998.

**Ulaby FT, Moore RK, Fung AK. Microwave Remote Sensing: Active and Passive. Volume 1 - Microwave Remote Sensing Fundamentals and Radiometry. Artech House Publishers; 1981.

**Response to Reviewer #3 (Dr Mohammed Shokr)**

**General Comments:**

*The manuscript introduces a new algorithm to retrieve sea ice concentration in the Arctic region based on radiometric observations from the L-band SMOS, with its multi-viewing angles. It uses the Maximum Likelihood criterion with input distributions of a couple of radiometric indices, assuming Gaussian distributions with mean and standard deviation obtained from model and observation findings.*

*The idea is interesting, the study is scientifically warranted and it is an appropriate continuation, with a remarkable progress, of the limited number of similar studies using SMOS in the past 8 years. I would recommend publication and certainly commend the authors on their effort. However, one should keep in mind that the challenge of estimating SIC from remote sensing data remains.*

*In most studies, sea ice is treated as one entity. Yet, in nature it is manifested in several types with large diversity of physical properties that may lead to consider it as different entities. Thin ice, seasonal ice, perennial ice, summer ice, etc. are different "creatures" within the realm of sea ice; let alone snow-covered ice under different metamorphosed snow conditions that lend itself to different radiative properties, even under the passive L-band radiometry. Given this concept, I would be cautious when considering SIC algorithms trying to approximate the wide range of ice properties into a single set.*

**Authors:** **We thank the reviewer for his summary and praise of the study.**

*With this in mind I am not comfortable with the title of the paper that groups all ice types under one phrase "sea ice in the Arctic Ocean", but I guess nothing can be done here! But it would be more appropriate to use the word "Estimating" in the title instead of "Measuring" since we don't really measure SIC.*

**Authors:** **Changed the title from Measuring to Estimating.**

*The method adds to the tools of estimating ice concentration from microwave data. One advantage of having different methods is to be able to perform cross-checking. This is important because there is no reliable "truth" data against which we can evaluate each method. All methods suffer from errors and the only way to approach the "truth" is by cross-checking. For example, the current study performs the validation by comparing results against maps from OSI-SAF. But the latter, much like any other operational ice maps, may not be considered truth data either.*

**Authors:** We fully agree that the method presented here will enable independent evaluation of results obtained using other methods. Cross-checking of results is a tenet of the scientific method.

*Another concern is about a sentence in the Abstract; "We find that sea ice concentration is well determined (correlations of about 0.75) when compared to estimates from other sensors such as the Special Sensor Microwave/Imager (SSM/I and SSMIS)." Retrieval of any parameter from remote sensing data is associated not only to the sensor characteristics but also to the retrieval method. I believe the method used in this study is new, so if there are other method using L-band then the authors can do comparison. To keep the above statement, the authors should qualify it; namely to say correlate well with other passive microwave – but under what condition (when and where). I think it should correlate well with other sensors over mature Arctic sea ice in winter. Other than that I don't think the correlation would reach 0.75.*

**Authors:** We have changed the sentence to be more explicit about the correlation, now specifying both the elements involved in the comparison (i.e., MLE/SMOS vs OSI-SAF/SSMIS) as well as the span both in the time (one year, 2014) and space (pan-Arctic) domains. The *changes are done in lines 9-14, page 1 from the track change version of the manuscript.*

To the best of our knowledge, this is a completely new approach to SIC estimates in that this is the first time that L-band observations are used to estimate SIC and also the first time that a MLE criterion is used to retrieve SIC.

**Specific Comments:**

*A few suggestions for corrections are listed below. It would be nice if the authors consider them while preparing the final submission.*

*Page 4 Line 21: correct the sentence to be "Hereafter we will use TB to refer to surface brightness temperature, for simplicity"*

**Authors:** Done, *line 140, page 5.*

*Page 4 Line 25: correct the sentence to be "is the ratio between reflected and incident radiation"*

**Authors:** Done, *line 144, page 5.*

*Page 5 Line 4: no need to mention the refractive index (n); this is for optical remote sensing but here we deal with microwave.*

**Authors:** **Done,** *line 145, page 5.*

*Page 5 line 6: the sentence "The nonlinearity is an advantageous property for remote sensing ..." is not explained. How? Also it has no relevance to the text before it. The authors may remove it.*

**Authors:** **Removed,** *line 155, page 5.*

*Page 5 Line 7: if Fig. 2 is for the L-band please mention that in this line or in the figure caption. Also, while the authors mentioned the seawater and sea ice parameters that are used in equation 3, they did not mention the snow parameters. Here they have to be careful because it is difficult to characterize the snow by a single temperature value as it is highly responsive to the air temperature. Even for dry snow, it can be lossy because the salinity at the snow base is usually higher than 0; it can be as high as 20 ppt or higher as shown in many studies.*

**Authors:** **Changed both the text and figure caption to indicate this is L-band, and also added the snow values used.**

**On the snow comment, Schwank et al 2015 confirms that at L-band the imaginary part of the dielectric constant is very small compared with the real part, and stats that it can be neglect in the model of snow dielectric constant. This has been clarified in the manuscript in lines 182-190, page 6.**

*Page 5 line 23: "dry snow still has an effect in (make it "on" not "in") emissivity that changes with the angle of incidence according to Snell's law". Snell's law is about the angle of refraction, nothing to do with the emissivity.*

**Authors:** **Thanks, we have modified the text accordingly, in lines 189, page 6**

*Page 6: in the set of presented equations I think one equation is missing; that is the one that determines the reflectivity from the ice surface in terms of its dielectric constant (i.e. Fresnel equation). This should be inserted before equation 5.*

**Authors:** **Inserted, see Eq. (3) in page 5**

*Page 7 line 7: close the bracket after (resulting from ....unknown physical parameters).*

**Authors:** **Done**

*Page 7 line 10: something wrong in the sentence "... among conditions such as deletedwhen a phase change ..."*

**Authors:** **Done**

*Page 7: just wonder why didn't you use the polarization ratio instead of the polarization difference? The former is more common and it eliminates the dependence of the brightness*

*temperature of the physical temperature. I am not suggesting to change the present scheme but an inclusion of a sentence to explain why PD and not PR would be useful.*

**Authors: We have preferred to use PD instead of PR, since we have verified that the former has a larger dynamic range than the latter. Given the SMOS measurement errors a parameter like PD with a relatively large dynamic range is more suitable for SIC retrieval purposes. This is now clarified in the text in lines 278-280, page 9.**

*Page 7 Line 25 "we will use tie points as ground truth estimates of sea ice concentration". What does that mean? Tie points are used to estimate ice concentration based on a set of algebraic equations.*

**Authors: Modified in line 284 page 9.**

*Page 8 Line 7: just a comment on Figure 5, it is good to see the model confirms what we know – that Tb, not PR or PD can be used to estimate ice thickness. The latter are good for estimating ice concentration.*

**Authors: Good remark, we have added a sentence to that effect "Compared with TB, the total variation of both AD and PD with ice thickness are significantly smaller and, therefore, are better suited to estimate ice concentration" in line 209 page 10.**

*Page 9 Line 3: the statement "AD is the most robust index to retrieve SIC, slightly better than PD, and significantly better than TB, as TB is highly sensitive to ice thickness variations" may need some more thought. The fact that TB in the L-band is sensitive to the thickness has no relevance to its robustness in retrieving SIC of total ice (i.e. concentration regardless of ice thickness), hence the above statement may not be accurate. As mentioned in the text, the L-band has problem in estimating SIC only when the ice is thin (a few centimeter) and becomes partially transparent to the L-band. I think what can be used to comment on the high value of the propagated error in Table 3 when TB is used is the fact that the variability of TB is quite high with the thickness parameter (unlike the temperature and salinity) because of the large penetration of the L-band, and may not conform to the Gaussian assumption.*

**Authors: Thanks, this is a very interesting remark. We have introduced a sentence on the text to account for this, in line 338 page 11.**

*Page 12 Line 1: change the word "replacedmoenthsepochs".*

**Authors: Done.**

*Page 12 Line 1: when talking about Fig. 10, the given information is expected, nothing new. I would prefer seeing Fig. 10 generated for data in winter months. Ice in Laptev and Kara seas remains thin during winter. The region remains a marginal ice zone throughout most of the*

*freezing season. So, I believe that we will see the same difference during November and December as we see it during the period 2-5 November shown in Fig. 10. But it is interesting to confirm that. The authors may refer to a publication about thin ice in the Arctic titled "Interannual variability of young ice in the Arctic estimated between 2002 and 2009".*

**Authors: The purpose of Figure 10 is to show that the main difference between using the AD index vs the AD plus PD indices occurs in regions known to be covered with thin ice, not (at least yet) to turn it around into a possible analysis tool for locating regions of thin ice. We have added the proposed reference and a sentence explaining the situation observed by the reviewer (lines 502-507, page 16).**

**However, we add the figures the reviewer is asking for, for his information:**

[Figure]

*Page 13 Line 5: Same argument applies here. The text says "... for some days in November, the month of maximum extension of thin young ice". My argument is that this statement applies to ice extending west through the Beaufort Sea" but not in the Laptev and Kara Seas area, where thin ice cover continues to exist in winter.*

**Authors: A comment and the reference have both been added.**

*Figure 6: in the caption it should be mentioned that the values for the ice are coming from multi-year ice (as mentioned in the text).*

**Authors: Done.**

*Figure 10: I believe that the difference of SIC is presented in scale of tenth concentration. Please indicate that in the caption. The advantage of this figure is not limited to what is already described in the paragraph. It also marks the area of highly dynamic and ice reproduction, which, again not limited to November.*

**Authors: Agreed and done.**

*Finally - the phrase "tie-point regions" is confusing. Better use "regions for generating tie-points"*

**AUTHORS: Agreed and done.**

**Response to Reviewer #4 (anonymous)**

**General Comments:**

*The authors develop a new algorithm to retrieve sea ice concentration from the L-Band SMOS measurements at 1.4 GHz. At 1.4 GHz, the influence of the atmospheric properties on brightness temperatures is very low and, additionally, SMOS provides full-polarized measurements at different incident angles. Due to a higher penetration depth, information about the sea-ice thickness can be retrieved in addition to sea-ice concentration. Ideally, the method here is a first step to combining sea-ice concentration (SIC) and sea-ice thickness measurements from the same place at the same time.*

*For their new SIC retrieval method, the authors take advantage of the special features provided by SMOS and introduce two indices, the polarization difference and the angular difference, to avoid the dependence of the brightness temperature on the sea-ice thickness. They use a Maximum Likelihood Estimator in combination with these two indices in opposite to the more usual method of linear estimation used in algorithms designed for higher frequencies. They find that the retrieved SIC compare well with observations, except in fall, where there are differences in regions of thin ice due to the high penetration depth of the low-frequency radiation.*

*The topic is timely and the approach of the Maximum Likelihood Estimator is interesting. The authors took well into account previous comments and therefore improved the manuscript notably. However, I think there is still room for improvement in the structure and writing style. The clarity of your message would profit from a careful structure- and writing-oriented (instead of topic-oriented) proof-reading.*

*I suggest minor revisions. I have some comments and questions and I hope the authors can answer them. Also, I have some suggestions that could improve the clarity of the manuscript.*

**Authors: We thank the reviewer for his/her comments have helped improve the manuscript.**

**Thematic comments:**

*#1 It is not totally clear to me what is the advantage of this new method, with which I mean a SIC retrieval at 1.4 GHz. I can understand that there has not been any SIC retrieval at this frequency before but is it then not only one new method amongst others to retrieve SIC? If I understood right, this retrieval yields smaller errors in summer. You could underline a bit more that this is a*

*key advantage compared to other algorithms, which have problems in summer due to melt ponds and wet snow for example.*

**Authors:** We feel the manuscript already addresses some key advantages of using L-band observations, which Fig. 1 captures visually, and an MLE optimization approach. These include the negligible effect of the atmosphere on the L-band measurements, its lesser sensitivity to temperature changes relative to radiometers that operate at higher frequencies, or the lack of any significant effect of snow depth on Tb V-pol measurements. Furthermore, Reviewer#3 made a valid point, the prospect that a new independent method affords to cross-check SIC results obtained with other methods. Moreover, as rightly pointed out by this reviewer, we expect this method to be better suited to SIC estimation during the wet summer months than others that are not based on L-band measurements. Quantifying this last statement would however require a comprehensive analysis including the construction of a match-up, quality-controlled database of optical imagery. This is beyond the scope of this study, and we leave it for future work.

**A discussion on that point is added in the Conclusions section, in lines 627-635, page 19.**

*#2 On the same note, you state as an advantage that, as we now could in principal retrieve both sea-ice thickness and SIC from 1.4 GHz-measurements, these could be combined to retrieve both at the same time. But can thickness and concentration be retrieved at the same time if the retrieval method for SIC has problems in the thin ice areas (under 60 cm) and the retrieval method for sea-ice thickness is only for thin ice areas, up to 50 cm (Kaleschke et al., 2012; Huntemann el al., 2014)? I would like the authors to comment on that.*

**Authors:** It feels that it should be possible to estimate SIC and SIT simultaneously thanks to the multi-angular observational feature of SMOS and the availability of two polarization. Although we had originally included this statement to motivate such a study, we have now removed it to only present the facts, and to steer clear from any controversy that an unsubstantiated hypothesis might carry. Still, we believe this would be a very important result, and we plan to pursue such study in the future.

**Writing comments:**

*#3 The words "below", "above", "former", "latter" are used a lot. Often this is too vague and it is not clear what exactly is meant by them. The reader is pointed in a lot of directions and gets off the track of the actual message. I suggest that you rethink the structure of the manuscript to avoid as much as possible having to point to another place in a manuscript.*

**Authors**: We thank the reviewer for his/her stylistic critique, to which we have tried to accommodate by forgoing those adverbs and adjectives when inessentials, and in several

**other ways. We hope that the style and flow of the manuscript has thus improved, and that grammatical errors have also been resolved. And we obviously leave for TC copy-editing to ensure its consistency with the overall TC style.**

*#4 It is sometimes unclear what was done in the study and what has been done before. As you use several tenses (past, perfect, present, future) and active and passive mode in an inconsistent way (sometimes changing in the middle of a paragraph), I suggest to carefully proof-read the manuscript and and to correct the inconsistencies. This would remove some of the confusion.*

**Authors: Thank you, ditto.**

*#5 Also, very often a paragraph or sentence starts with "the figure shows", "the table shows" or "the authors show". I think if the emphasis was on the message of the figure/reference and the figure/reference was only given in parenthesis at the end of the sentence, your message would gain in clarity.*

*Consider as an example the difference between:*

*P13 L1-4 : "Figure 11 shows the spatial distribution of SIC in the Arctic Ocean estimated from (a) SMOS for the 3-day period 2–5 March 2015, (b) OSI-SAF SIC on 4 March 2014, and (c) the difference between (b) and (a). "*

*P14 L22-23 : "However, the sensitivity of the brightness temperature to sea surface temperature, atmosphere, and wind speed is clearly reduced when observing the sea surface with radiometers working at lower frequencies (Figure 1) ..."*

**Authors: Ditto. However, besides the stylistic approach, the level of complexity that figures convey can be markedly different. Some, like Figures 11, require an introduction to what is being shown before the message(s) can be given while others, like Figure 1, are more of the single-sentence message type.**

*#6 I find it confusing when parentheses are used for a whole sentence. Either it is important for the study, then the sentence can be written as such, or it is not important and it can be left out.*

**Authors: Ditto.**

*#7 Some sentences are very long with several dependencies and often too many or too less commas. I lost track several times. Shorter sentences would improve clarity.*

**Authors: Ditto.**

*#8 There are still several typos and grammatical mistakes. I tried to highlight some of them but I suggest that you let a native speaker or just another person read through your manuscript.*

**Authors:** We thank again the reviewer for his/her careful review and comments. We have tried our best to accommodate them, and we hope as a result the readability of the manuscript has improved.

**Specific Comments:**

*P1 L11: I don't see the logical connection from the previous sentence to the "therefore"*

**Authors: Done**

*P1 L18: Replace "ice cover" by "sea-ice cover"*

**Authors: Done, in line 25, page 2**

*P2 L5-7: The sentence is unclear as you use both "therefore" and "because". Try to divide it in to two sentences.*

**Authors: Done, lines 49-45, page 2**

*P2 L7 : Replace "since" by "for"*

**Authors: Done**

*P2 L16 : Comma after "that" and it is not clear to what "former" refers. Add "the ice penetration of the former".*

**Authors: Done, lines 56, page 2**

*P2 L22: I think you can leave out "what is left for a future work"*

**Authors: Done**

*P2 L29: Not clear why the spatial resolution of 35 to 50 km is a key feature. Maybe remove "key" in L28.*

**Authors: Done, line 71, page 3.**

*P3 L10: Remove the sentence in parentheses.*

**Authors: Done**

*P3 L30-31: Move the product version to the OSI SAF parenthesis.*

**Authors: Done**

*P4 L3 and L7 : I think you can remove "see below".*

**Authors:** Done

*P4 L5 : You always only mention data from 2014. Would it be right to only write "from the year 2014"? And then you could leave out in the rest of the manuscript all the time you mention "over the year 2014". If you use more years (that did not become clear to me), write "from 2014 to XXXX". Otherwise it is not clear when your data period ends.*

**Authors:** Done, line 125, page 4.

*P4 L13 : Remove "As discussed in Section 1"*

**Authors:** Done

*P4 L13 : Replace "different incidece angle at" by "different incidence angles to"*

**Authors:** Done

*P4 L14: I think you can start a new sentence: "TB can be expressed as".*

**Authors:** Done

*P4 L18: Replace "into" by "on"*

**Authors:** Done

*P4 L21: Write sentence without parentheses. Do you mean "We use TB to refer to surface brightness temperature"? The surface emissivity would be e_s, right? This is not clear.*

**Authors:** Done, line 144, page 5.

*P5 L4-5: There is no logical sequence between "varies linearly with emissivity" and "The nonlinearity".*

**Authors:** Done, we have deleted the nonlinearity sentence, line 155, page 5.

*P5 L7: Replace "with" with "on the"*

**Authors:** Done

*P5 L8-10: Write the sentence without parentheses*

**Authors:** Done. In fact the sentence has been changed considerably, line 145, page 5

*P5 L20: I think you can remove the sentence in parentheses.*

**Authors:** Done

*P5 L30: I think it would be clearer to introduce the equation directly when you cite it for the first time (near L8). Maybe it would work if you change the sequence of the paragraphs on P5 and P6.*

**Authors:** **Done. Now Eq. 5 is not cited before Eq. 4. We prefer not put Eq. 4 when we cite in the first time (line 174. Pg 6), since other assumptions and terms need to be defined before presented this equation.**

*P6 L25-29: I think this sentence is too long.*

**Authors:** **These are several sentences. We don't think that cutting it is necessary.**

*P7 L2: I think "microwave remote sensing model" is not the right term here. Maybe you mean "microwave emission model"?*

**Authors:** **Correct, done in line 254, pg 8.**

*P7 L10: Remove "deletedwhen"*

**Authors:** **Done**

*P7 L9-11: I don't understand this sentence.*

**Authors:** **Removed, it was not essential and was adding confusion.**

*P7 L20: It is not clear what "the former" refers to.*

**Authors:** **Indeed, now clarified as "the V-pol", Line 274, page 9.**

*P7 L24-29: "In this study","in this context","in this applications". The logical sequence of these sentences is not clear.*

**Authors:** **The three sentences have been rephrased to improve readability (see Line 285-290, pàge 9)**

*P8 L1: you could cite (Tab. 1) after ice*

**Authors:** **Done (line 294, page 9)**

*P8 L6: "indicated above" is not specific enough, it could be 25°, 60°, 30°, 50°.*

**Authors:** **Clarified as sigma = 50º and sigma = 25º, in line 300, page 10.**

*P8 L28: I don't know if "reasonable" is the right word here. Maybe "average"?*

**Authors:** **Reasonable is a better description, average would not be correct.**

*P9 L5: "done by other authors". It is not clear to what "done" refers. Did they focus on TB or on inversion algorithms using PD and AD? I am quite sure that it is the former but this is not clear from the sentence structure.*

**Authors: Indeed, they focussed on TB. But, in the new version the reference has been deleted, since it has been already explained before. (line 344, page 11)**

*P9 L14: If the level of uncertainty is unquantified, does it still mean that it is negligible?*

**Authors:  No, it is not negligible, but it means that the error is not known. We have removed the word "unquantified." (line 354, page 11)**

*P9 L27: add "defined" before "above"*

**Authors: Done**

*P9 L32-P10 L2: The paragraph starts with "Table 1 lists". It reads as if you were introducing something new. But, actually, you have referenced Table 1 several times before. This relates again to the structural issue (see #3).*

**Authors:  Edited the sentence (line 378, page 12). However, the focus of this sentence is different from the sentences in live 358 which also cite Table 1.**

*P11 L5: If possible, try to use another letter for the distributions. rho was already used for the density before.*

**Authors:  The reviewer is right. We have changed rho by f in Eqs. 15 and 16.**

*Section 5.1.: This is more a listing of different figures than a coherent story (see #5). I suggest that you rethink about the message you want to convey in this section.*

**Authors: Section edited for improve readability and flow.**

*P12 L1: Remove "replacedmoenths" and replace "epochs" by "periods".*

**Authors: Done**

*P12 L29: I think you can remove "As we have shown"*

**Authors: Done in line 480, p 15.**

*P13 L5: I think you can remove "here" and "some days in"*

**Authors: Done**

*P13 L9: It is not clear to what "that response" relates to.*

**Authors:** Done in line 507, page 16.

*P13 L31-33: Reformulate this sentence. Not clear*

**Authors:** Done in 529, page 16.

*P14 L6: I think you can replace "that is, the square of correlation coefficients" by "R^2" or "r^2"*

**Authors:** Adopted in 567, page 17.

*P14 L11: You don't need to put parentheses here*

**Authors:** Done

*P14 L19: It is not clear if this has been identified by Ivanova et al. (2015) or by someone else.*

**Authors:** Changed for clarification.

*P14 L30: You introduce the full names of AD and PD only later in the conclusions (P15 L6).*

**Authors: The acronyms PD and AD are defined where they first appear, in Sect. 4.1. They had been re-defined here to make the conclusion self-contained. We have no preference, leaving it to TC style.**

*P15 L24: Do you mean "determination" or "correlation" coefficients?*

**Authors: Determination.**

**Figures:**

*The caption often contains more explanations than are needed. And I think you do not need to reference in which part of the text the figure is discussed. So I think you can remove all "see XXX".*

**Authors: Done.**

*Figure 1: A similar sentence can be found in the text, I think you can remote it*

**Authors: Done.**

*Figure 10, 11 and 12: I suggest that you use different colorbars. This would improve the clarity of the figures. You could use blue-white-red for difference plots and blues or reds for absolute values.*

**Authors:** **We have modified the color palettes of difference figures to improve highlighting of features, but we consider that our choice is quite appropriate for absolute value.**

*Figure 11 and 12: Replace "3th" by "3rd"*

**Authors:** **Done.**

*Figure 14: In the colorbar, replace "SAF<0.9" by "Both<0.9".*

**Authors:** **Done.**

**Authors:** **All the references have been reviewed and corrected.**

[revised manuscript text omitted]

---

## Author Response (AR3)

17 July 2017

Prof. Lars Kaleschke
Scientific Editor
*The Cryosphere*

**Manuscript tc-2016-175**

Dear Prof. Lars Kaleschke,

Please find attached the revised manuscript entitled *Estimating sea ice concentration in the Arctic Ocean using SMOS.* (Please note the title change.) This manuscript has been revised and improved based on the comments we received from you.

We also attach the new version of the manuscript with marked changes (in blue color) with respect to previous version. Below we describe the revisions made to the manuscript and the response to the editors' comments.

Sincerely,

Dr. Carolina Gabarro
Barcelona Expert Center
Physical Oceanography Department
Marine Sciences Institute (ICM)
National Research Council (CSIC)
E-mail: cgabarro@icm.csic.es

**Response to the Editor (Prof. Lars Kaleschke)**

(*Original comments are shown in italics*; **Authors: responses in boldfaced font; number of lines and pages is from the track changes version or the manuscript.**)

Minor changes/suggestions:

*The title is very general and does not reflect the method & parameters used, area/region and period of investigation.*

**Authors: The new title we propose is: 'New methodology to estimate Arctic sea ice concentration from SMOS combining brightness temperature differences in a maximum likelihood estimator'. We hope this follows the editor's recommendations.**

**Editor: The abstract shall contain all major findings, e.g. about the assessment of used parameters, and the main motivation.**

**Authors: We have changed completely the abstract to:**

**Monitoring sea ice concentration is required for operational and climate studies in the Arctic Sea. Technologies used so far for estimating sea ice concentration have some limitations, as for instance the impact of the atmosphere, the physical temperature of ice, the presence of snow and/or melting, etc. In the last years, L-band radiometry has been successfully used to study some properties of sea ice, remarkably sea ice thickness. However, the potential of satellite L-band observations for obtaining sea ice concentration had not yet been explored.**

**In this paper, we present preliminary evidence showing that data from Soil Moisture Ocean Salinity (SMOS) mission can be used to estimate sea ice concentration. Our method, based on a Maximum Likelihood Estimator (MLE), exploits the marked difference in the radiative properties of sea ice and seawater. In addition, the brightness temperatures of 100% sea ice and 100% sea water, as well as their combined values (polarization and angular difference), have been shown to be very stable during winter and spring, so they are robust in front of variations in physical temperature and other geophysical parameters. Therefore, we can use just two set of tie points, one for summer and another for winter, for calculating sea ice concentration, leading to a more robust estimate.**

**After analysing the full year 2014 in the entire Arctic, we have found that the sea ice concentration obtained with our method is well determined as compared to Ocean and Sea Ice Satellite Application Facility (OSI-SAF) dataset. However, when thin sea ice is present (ice thickness <0.6 m) the method underestimates the actual sea ice concentration.**

Our results open the way for a systematic exploitation of SMOS data for monitoring sea ice concentration, at least for specific seasons. Additionally, SMOS data can be synergistically combined with data from other sensors to monitoring of pan-Arctic sea ice conditions.

*Editor:* *First section of conclusion reads more like an introduction, not conclusion. Some key messages are missing in the conclusion, e.g. the advantage (better quality) of AD over the combination of PD & AD as previously stated in the discussion.*

**Authors:** **We have moved two sentences to the introduction section and added two new sentences in the Conclusions. The added sentences are:**

Two indices derived from brightness temperature, PD and AD, have been chosen, since they verify the two required conditions: they maximize the difference between open water and sea ice, and they present a low response to changes in the geophysical characteristics of the media.

We have shown that the best configuration for SIC retrieval is using AD only with the MLE inversion method. We exclude PD and TB because they are more sensitive to ice thickness; therefore, the combined use of AD and PD presents larger errors when thin ice is present (fall). The MLE inversion method presents better results than a linear inversion since it takes into account the uncertainty of the tie-points.

*Editor:* *Figure 10. Which date?*

**Authors:** **Yes sorry. This was an error, it is now added on the manuscript.**

[revised manuscript text omitted]